# Screening uptake of colonoscopy versus fecal immunochemical testing in first-degree relatives of patients with non-syndromic colorectal cancer: A multicenter, open-label, parallel-group, randomized trial (ParCoFit study)

Natalia González-López[1◦], Enrique Quintero[1,2◦]*, Antonio Z. Gimeno-Garcia[1,2], Luis Bujanda[3,4,5], Jesús Banales[3,4,5], Joaquin Cubiella[6], María Salve-Bouzo[6], Jesus Miguel Herrero-Rivas[6], Estela Cid-Delgado[6], Victoria Alvarez-Sanchez[7], Alejandro Ledo-Rodríguez[7], Maria Luisa de-Castro-Parga[8], Romina Fernández-Poceiro[8], Luciano Sanromán-Álvarez[8], Jose Santiago-Garcia[9], Alberto Herreros-de-Tejada[9], Teresa Ocaña-Bombardo[10], Francesc Balaguer[10], María Rodríguez-Soler[11], Rodrigo Jover[11], Marta Ponce[12], Cristina Alvarez-Urturi[13], Xavier Bessa[13], Maria-Pilar Roncales[14], Federico Sopeña[14], Angel Lanas[14], David Nicolás-Pérez[1], Zaida Adrián-de-Ganzo[1], Marta Carrillo-Palau[1], Enrique González-Dávila[15], On behalf of the Oncology Group of Asociación Española de Gastroenterología

1 Department of Gastroenterology of Hospital Universitario de Canarias, La Laguna, Tenerife, Spain, 2 Instituto Universitario de Tecnologías Biomédicas (ITB) & Centro de Investigación Biomédica de Canarias (CIBICAN), Universidad de La Laguna, La Laguna, Tenerife, Spain, 3 Department of Gastroenterology of Hospital Universitario Donostia, Instituto Biodonostia, Centro de Investigación Biomédica en Red de Enfermedades Hepáticas y Digestivas (CIBERehd), Universidad del País Vasco (UPV/EHU), San Sebastián, Spain, 4 IKERBASQUE, Basque Foundation for Science, Bilbao, Spain, 5 Department of Biochemistry and Genetics, School of Sciences, University of Navarra, Pamplona, Spain, 6 Department of Gastroenterology, Hospital Universitario de Ourense, Ourense, Spain, 7 Department of Gastroenterology of Complexo Hospitalario de Pontevedra, Pontevedra, Spain, 8 Department of Gastroenterology of Complexo Hospitalario de Vigo, Vigo, Spain, 9 IDIPHISA, Department of Gastroenterology of Hospital Universitario Puerta de Hierro-Majadahonda o, Madrid, Spain, 10 Department of Gastroenterology, Hospital Clínic Barcelona, Centro de Investigación Biomédica en Red en Enfermedades Hepáticas y Digestivas (CIBEREHD), IDIBAPS (Institut d'Investigacions Biomèdiques August Pi i Sunyer), University of Barcelona, Barcelona, Spain, 11 Department of Gastroenterology, Instituto de Investigación Sanitaria ISABIAL, Hospital General Universitario Dr. Balmis, Departamento de Medicina Clínica, Universidad Miguel Hernández, Alicante, Spain, 12 Department of Gastroenterology of Hospital Universitario La Fe de Valencia, Valencia, Spain, 13 Gastroenterology Department, Hospital del Mar Medical Research Institute (IMIM), Barcelona, Spain, 14 Department of Gastroenterology of Hospital Universitario Lozano Blesa de Zaragoza, IIS Aragón. CIBERehd, Zaragoza, Spain, 15 Departamento de Matemáticas, Estadística e Investigación Operativa, Instituto IMAULL, Universidad de La Laguna, San Cristóbal de La Laguna, Spain

◦ These authors contributed equally to this work.
* equinter@ull.edu.es

**Data Availability Statement:** All relevant data are within the manuscript and its Supporting Information files.

## Abstract

### Background

Colonoscopy screening is underused by first-degree relatives (FDRs) of patients with non-syndromic colorectal cancer (CRC) with screening completion rates below 50%. Studies

**Funding:** This work was granted by Instituto de Salud Carlos III (ISCIII). Spanish Government (FIS PI15/01257 to AZG) (http://www.imib.es/ServletDocument?document=24268). The funders had no role in study design, data collection and analysis, decision to publish, or preparation of the manuscript.

**Competing interests:** EQ and AL received an honorarium for consultancy from Sysmex (2017–2020). FB received endoscopic equipment on loan of Fujifilm, received an honorarium for consultancy from Sysmex (2017–2020) and editorial fee from Elsevier as editor of Gastroenterologia y Hepatologia. The other authors declare no conflict of interest regarding this study.

**Abbreviations:** CI, confidence interval; CRC, colorectal cancer; FDR, first-degree relative; FIT, fecal immunochemical testing; OR, odds ratio; WHO, World Health Organization.

conducted in FDR referred for screening suggest that fecal immunochemical testing (FIT) was not inferior to colonoscopy in terms of diagnostic yield and tumor staging, but screening uptake of FIT has not yet been tested in this population. In this study, we investigated whether the uptake of FIT screening is superior to the uptake of colonoscopy screening in the familial-risk population, with an equivalent effect on CRC detection.

## Methods and findings

This open-label, parallel-group, randomized trial was conducted in 12 Spanish centers between February 2016 and December 2021. Eligible individuals included asymptomatic FDR of index cases <60 years, siblings or ≥2 FDR with CRC. The primary outcome was to compare screening uptake between colonoscopy and FIT. The secondary outcome was to determine the efficacy of each strategy to detect advanced colorectal neoplasia (adenoma or serrated polyps ≥10 mm, polyps with tubulovillous architecture, high-grade dysplasia, and/or CRC). Screening-naïve FDR were randomized (1:1) to one-time colonoscopy versus annual FIT during 3 consecutive years followed by a work-up colonoscopy in the case of a positive test. Randomization was performed before signing the informed consent using computer-generated allocation algorithm based on stratified block randomization. Multivariable regression analysis was performed by intention-to-screen. On December 31, 2019, when 81% of the estimated sample size was reached, the trial was terminated prematurely after an interim analysis for futility. Study outcomes were further analyzed through 2-year follow-up. The main limitation of this study was the impossibility of collecting information on eligible individuals who declined to participate.

A total of 1,790 FDR of 460 index cases were evaluated for inclusion, of whom 870 were assigned to undergo one-time colonoscopy (*n* = 431) or FIT (*n* = 439). Of them, 383 (44.0%) attended the appointment and signed the informed consent: 147/431 (34.1%) FDR received colonoscopy-based screening and 158/439 (35.9%) underwent FIT-based screening (odds ratio [OR] 1.08; 95% confidence intervals [CI] [0.82, 1.44], *p* = 0.564). The detection rate of advanced colorectal neoplasia was significantly higher in the colonoscopy group than in the FIT group (OR 3.64, 95% CI [1.55, 8.53], *p* = 0.003). Study outcomes did not change throughout follow-up.

## Conclusions

In this study, compared to colonoscopy, FIT screening did not improve screening uptake by individuals at high risk of CRC, resulting in less detection of advanced colorectal neoplasia. Further studies are needed to assess how screening uptake could be improved in this high-risk group, including by inclusion in population-based screening programs.

## Trial registration

This trial was registered with ClinicalTrials.gov (NCT02567045).

## Author summary

### Why was this study done?

- The risk of colorectal cancer (CRC) is 3 to 4 times higher in first-degree relative (FDR) of patients with non-syndromic CRC. These individuals are considered candidates for colonoscopy-based screening starting at 40 years of age, but this approach is associated with a suboptimal acceptance rate of approximately 50%.

- Recent evidence suggests that annual fecal immunochemical testing (FIT) may be equivalent to colonoscopy for detecting CRC and advanced adenomas, but the acceptance of this strategy is unknown in the familial-risk population.

- This study was designed to test the hypothesis that the uptake of FIT screening is superior to the uptake of colonoscopy screening in this population, with an equivalent effect on CRC detection.

### What did the researchers do and find?

- This multicenter randomized controlled trial included 870 asymptomatic FDR of patients with non-syndromic CRC. Participants were invited to participate through their affected index case(s).

- FDR were randomized (1:1) to one-time colonoscopy versus annual FIT during 3 consecutive years followed by a work-up colonoscopy in the case of a positive test.

- The rate of screening completion was similar in the group assigned to FIT and the group assigned to colonoscopy screening (36% versus 34%, respectively).

- The detection rate of advanced colorectal neoplasia was significantly lower in subjects receiving annual FIT than in those assigned to receive one-time colonoscopy.

### What do these findings mean?

- The findings of this trial indicate that FIT does not have the capacity of increasing the acceptance of screening in the non-syndromic familial CRC population.

- The fact that over 50% of eligible individuals refused to participate indicates that novel educational measures should be implemented to improve the awareness of individuals at risk and their providers.

- Future studies are needed to assess whether screening uptake can be improved for these individuals through their inclusion in population-based screening programs or by offering a choice between FIT and colonoscopy screening. The main limitation of this study was that it was not possible to collect information on eligible individuals who declined to participate, thus impeding our understanding behind low screening uptake in this population.

## Introduction

Family history and older age are the most important risk factors in colorectal cancer (CRC) development. The risk of developing CRC is almost doubled among first-degree relative (FDR) of patients diagnosed with CRC over the age of 60 years old. However, it is increased up to 3 to 4 times if the index case is younger than 60 years, and if there are siblings or 2 or more FDR affected in the family, regardless of age at diagnosis [1,2]. In addition, the risk of developing advanced colorectal neoplasia, a term that includes adenomas or serrated polyps ≥10 mm, polyps with tubulovillous architecture, high-grade dysplasia, and/or CRC, has been reported to be nearly doubled in individuals who had 2 FDRs diagnosed with CRC, compared to those with a single FDR affected with CRC or with average-risk individuals [3].

Apart from heightened risk, a pooled analysis of 8 epidemiological studies has shown that most individuals with a family history of CRC do not have a worse prognosis than those with no family history [4]. Overall, these studies suggest that most individuals with familial CRC risk would not benefit of intensive colonoscopy surveillance and could be screened in the same way as the average-risk population. Nevertheless, the most extended strategy for these individuals remains colonoscopy every 5 years, starting at the age of 40 years or 10 years before the youngest case in their family [5–7].

Decision analytic modeling suggests that colonoscopy screening is cost-effective in this population, assuming a 100% participation rate [8]. However, population studies have found that less than 50% of FDR of patients with CRC have undergone at least 1 colonoscopy since the index case CRC diagnosis [9–12], and compliance with colonoscopy every 5 years is even lower [13], which questions the efficacy of screening to reduce CRC incidence and mortality in this population.

A meta-analysis of 12 studies assessed the performance of fecal immunochemical testing (FIT) to detect colorectal neoplasia in those with familial risk [14]. The study revealed that FIT had acceptable accuracy for detecting CRC with sensitivity and specificity of 86% and 91%, respectively. In addition, a prospective randomized trial comparing FIT and colonoscopy in this population showed that annual FIT was equivalent to colonoscopy for detecting CRC or advanced adenoma [15]. However, the screening uptake of FIT in individuals at high-risk for familial CRC has not been analyzed yet. Therefore, this study was designed to test the hypothesis that uptake of FIT screening, followed by a work-up colonoscopy in the case of a positive test, is superior to the uptake of colonoscopy screening, with an equivalent effect on CRC detection, in the population with high familial risk.

## Methods

### Ethics statement

The Clinical Research Ethics Committee of Hospital Universitario de Canarias approved in writing the study protocol (S1 Text). All participants in the study provided written informed consent following randomization.

All authors had access to the study data and reviewed and approved the final manuscript.

### Study population

This randomized controlled trial was carried out in 7 Spanish regions (Aragón, Basque Country, Canary Islands, Cataluña, Galicia, Madrid, and Valencia), including 12 tertiary hospitals, between February 25, 2016 and December 31, 2021. Asymptomatic screening-naïve FDR (parents, siblings, and children) of index cases diagnosed with CRC during the previous 24

months were eligible if they met the following inclusion criteria: (a) having 1 index case younger than 60 years at the time of CRC diagnosis, having 2 or more index cases or a sibling with CRC, regardless of age at diagnosis; (b) age over 40 years or 10 years less than that of the youngest index case in the immediate family; (c) histological confirmation of CRC diagnosis of the index case; and (d) signing an informed consent form. Exclusion criteria included (a) previous CRC screening; (b) personal history of inflammatory bowel disease or colorectal neoplasia; (c) family history of hereditary CRC; (d) abdominal symptoms that required investigation; (e) previous colectomy; and (f) severe comorbidity that entailed a poor prognosis (average life expectancy less than 5 years). The trial was registered at ClinicalTrials.gov (number NCT02567045) and reported according to CONSORT (Consolidated Standards of Reporting Trials) [16].

## Selection process and screening invitation

At least 3 months after the diagnosis of CRC, and once oncologic surgery had been performed, index cases were contacted by phone to arrange an appointment at the high-risk CRC clinic office of each participating center. In the interviews, they were informed of the objectives of the study and were asked to sign an informed consent. If the index case met the inclusion criteria, a family tree of the first generation (parents, offspring, or siblings) was generated to identify all eligible and living relatives. At this point, an open-label randomization (1:1) was performed in FDR to undergo FIT for 3 consecutive years, and work-up colonoscopy if a positive test occurred, or straightforward colonoscopy. This was done using the randomization module in RedCap Electronic Data Capture (REDcap) [17]. Briefly, after initial data collection and verification of eligibility, stratified block randomization was carried following a pragmatic design (before signing the inform consent). The randomization process was independent to the investigators.

In the same interview, the index case received a personal invitation letter for each eligible family member and was asked to hand it to them. This letter described the importance of CRC prevention, provided a detailed written description of the aims, and included a formal invitation to participate in the assigned study group. A phone number was provided to the index case and eligible FDR to contact a member of the high-risk CRC clinic office at any time to schedule the enrollment appointment conveniently. Eligible individuals were given a leaflet with detailed information on the study outcomes, as well as the advantages and disadvantages of the assigned screening strategy. They were aware that they were part of a randomized study, and that colonoscopy is the standard approach for high-risk families. Those who did not attend the appointment were sent a reminder letter through the index case 2 months later.

## Study procedures

Colonoscopies were offered free of charge, including bowel-cleansing agents, and were performed by experienced endoscopists using the standard quality aspects defined by the Asociación Española de Gastroenterología [18]. Colon cleansing was performed as previously described [19]. The Boston bowel preparation scale was used for bowel cleansing assessment [20]. Colonoscopy was considered complete when the cleansing score was ≥2 points in each segment and the cecum was reached. Patients with an incomplete colonoscopy, due to any technical difficulty that impeded the exploration of the cecum, had to be evaluated by CT colonography or colonic capsule endoscopy.

Polyp size and morphology were recorded using the Paris classification [21]. Polyps were considered as advanced adenomas if they had size ≥10 mm, tubulovillous architecture, high-grade dysplasia, or in situ adenocarcinoma. Invasive CRC was considered when neoplastic

cells crossed the muscularis mucosae. Serrated polyps were classified according to the guidelines of the World Health Organization (WHO) [22]. Adenoma or serrated polyps ≥10 mm, polyps with tubulovillous architecture, high-grade dysplasia, and/or invasive CRC were grouped as advanced colorectal neoplasia.

Participants assigned to the FIT group received an annual automated quantitative FIT kit for 3 consecutive years with instructions for home use. They were notified to deliver it within 14 days after taking the sample. Individuals with ≥10 μg Hb/g feces were invited to undergo colonoscopy. We chose a low cutoff threshold because we wanted to increase the sensitivity of FIT for CRC detection. The individuals who did not deliver the test on time were contacted by telephone to offer them a new test.

Demographic data and CRC characteristics were recorded from the index cases. Epidemiological data from the eligible FDR included age, sex, history of colorectal neoplasia, substance abuse, comorbidity, deceased or untraceable relatives, and history of NSAID, aspirin, or anticoagulant treatment. Severe complications that occurred during colonoscopy or in the early termination of the procedure (immediate and delayed post-polypectomy hemorrhage and intestinal perforation) were also recorded. We considered severe post-polypectomy bleeding if prevented the conclusion of the procedure, transfusion or hospitalization was required. Intestinal perforation was defined as evidence of air, luminal contents, or instrumentation outside the gastrointestinal tract. The study data were collected and stored in REDcap, hosted at the Asociación Española de Gastroenterología, a database that guarantees data confidentiality [23].

## Outcomes

The primary outcome was to assess whether screening uptake of FIT was greater than that of one-time colonoscopy screening, in high-risk FDR of patients with non-syndromic CRC. Screening uptake was defined as the number of compliant participants divided by the number of eligible subjects in each screening strategy. In the FIT group, subjects were considered compliant if completed at least 1 FIT and the work-up colonoscopy if a positive test occurred. In the colonoscopy group, those subjects that underwent one-time colonoscopy were considered compliant. The secondary outcome was to determine the detection rate of advanced colorectal neoplasia in each group.

## Follow-up

Participants were actively followed from the last event registered before December 2019 until December 31, 2021. In the FIT group, an interval colonoscopy was defined as any colonoscopy performed after a negative FIT result. In the colonoscopy group, interval colonoscopy referred to colonoscopies performed within 36 months after a baseline colonoscopy. Unplanned FIT was defined as any FIT performed because of abdominal symptoms in both study groups or when it was performed as a screening tool in the colonoscopy group. Screening tests, interval colonoscopies, unplanned FIT, post-polypectomy surveillance, interval CRC, and deaths were identified through cross-linkage of the study database and the regional intranet network, which provides access to the electronic medical records at each site. Interval cancer was defined as cancer occurring between 6 and 36 months after a negative colonoscopy screening [24]. Lost to follow-up were considered FDR compliant or not during the inclusion period (February 2016 to December 2019), which had not additional information in their data records, at the regional intranet network, during the follow-up period (January 2020 to December 2021).

## Statistical analysis and sample size calculation

Screening uptake and detection rate of advanced colorectal neoplasia were assessed by intention-to-screen analysis. FDR who did not attend the initial appointment, and thus did not provide information about exclusion criteria, were considered as eligible and were included in the analysis. Individuals who did not comply with the assigned strategy were not allowed to change to the other group.

Between-group comparisons of the main outcome were calculated by multivariable logistic regression analysis with adjustment for FDR's age and sex, allocation of 1 or different screening strategies in the same family, index case tumor location, person (index case or other FDR) who attended the first appointment, and center (categorized as high or low recruiters, if they included more or less than 80 eligible individuals in the study, respectively). Results were reported as odds ratios (OR) with 95% confidence intervals (CI).

The detection rate of advanced colorectal neoplasia was the number of subjects with true positive results divided by the number of eligible subjects. The comparison of advanced colorectal neoplasia detection rate between the study groups was calculated by multivariable logistic regression analysis adjusting by the age of the FDR (categorized as having more or less than 54 years of age, according to the median age of participants), sex, and center.

Comparisons of continuous variables were performed using the Mann–Whitney U-test. Categorical variables with 2 categories were compared using the $\chi^2$ test. All analyses were performed using SPSS statistical software version 25.0.

The study was designed to achieve a 90% power and 95% confidence level for detecting an increase in the proportion of FDR undergoing CRC screening of 10% (from 50% in the colonoscopy group to 60% in the FIT group). According to these assumptions, and considering that up to 5% participants in each group would be lost to follow-up, the estimated sample size was 1,076 individuals (538 per group).

When 81% of the estimated sample size was reached, an interim analysis was conducted because recruitment was much lower than expected. Based on the screening uptake of 870 randomized FDR, the futility analysis [25] provided a conditional power of 2.95% a predictive power of 0.29% and a futility index of 97.1% (S1 Table). Therefore, on December 31, 2019, the trial Scientific Committee decided to interrupt the study for futility.

## Results

### Study population

Between February 2016 and December 2019, 460 index cases and 1,790 FDR were evaluated for inclusion in the study. Of these, 920 (51.4%) were not eligible. Overall, 870 FDR were randomized to undergo annual FIT ($n$ = 439) or colonoscopy ($n$ = 431) (Fig 1).

Table 1 shows the main demographic data from FDR and index cases. Kinship distribution was similar between groups, with siblings most frequently attending the initial appointment, followed by offspring and parents. Having 1 index case <60 years was the predominant inclusion criterion, followed by having a sibling and 2 or more FDR with CRC, regardless of age.

There were 604/870 (69.4%) FDRs who were randomly assigned to different strategies within the family, and 266/870 (30.6%) who were assigned to the same strategy or there was only 1 eligible FPG in the family (OR = 1.63, 95% CI [1.20, 2.22], $p$ = 0.001).

Table 2 shows the characteristics of relatives that agreed to participate but did not comply with the assigned strategy. Overall, there were 78 (20.3%) noncompliant participants, and 29/187 (15.5%) did not comply with the FIT strategy, whereas 49/196 (25.0%) declined to undergo colonoscopy in the colonoscopy group.

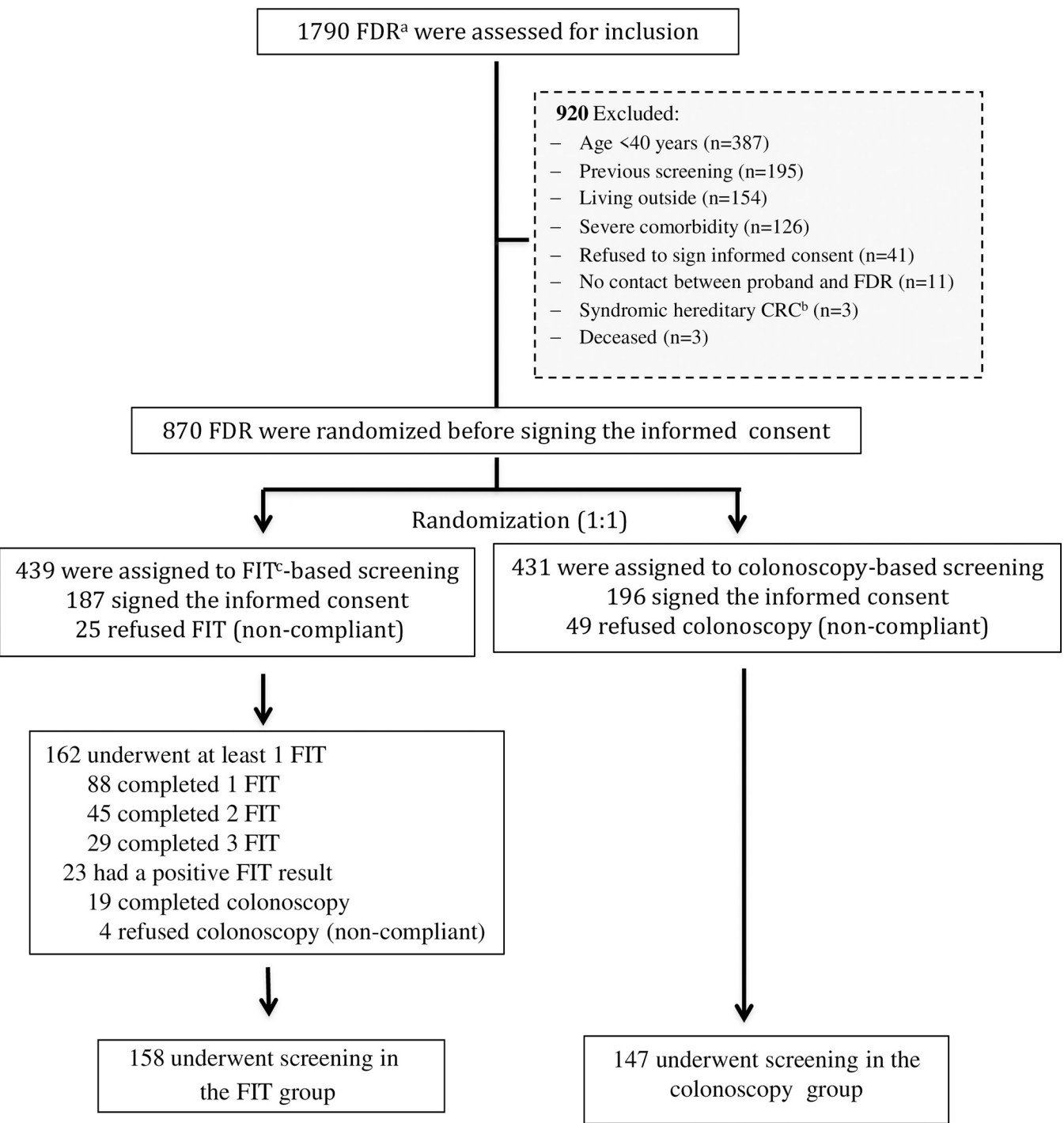

**Fig 1. Consort flow diagram (ParCoFit Trial).** [a]FDR = first-degree relative. [b]CRC = colorectal cancer. [c]FIT = fecal immunochemical test.

## Outcomes and follow-up

Among the 439 subjects assigned to FIT, 187 (42.6%) agreed to participate and 162 (36.9%) of them underwent at least 1 FIT: 88 (54.3%), 45 (27.8%), and 29 (17.9%) completed 1, 2, and 3 FIT, respectively. Overall, 158 (35.9%) were compliant with the FIT strategy, which included

**Table 1. Demographic data of FDR[a] and index cases included in the study.**

| Category | Colonoscopy group | FIT[b] group | Total |
|---|---|---|---|
| **Eligible population** | $N = 431$ | $N = 439$ | $N = 870$ |
| Male, $n$ (%) | 203 (47.1) | 216 (49.2) | 419 (48.2) |
| Female, ($n$%) | 228 (52.9) | 223 (50.8) | 451 (51.8) |
| Mean age ± SD | 55.9 ± 10.6 | 55.5 ± 10.3 | 55.7 ± 10.4 |
| **Participants** | $N = 196$ | $N = 187$ | $N = 383$ |
| Male, $n$ (%) | 100 (51) | 97 (51.9) | 197 (51.4) |
| Female, ($n$%) | 96 (49) | 90 (48.1) | 186 (48.6) |
| Mean age ± SD | 55.2 ± 10.1 | 54.1 ± 10.2 | 54.7 ± 10.2 |
| Age group, $n$ (%) | | | |
| <50 years | 63 (32.1) | 71 (38.0) | 134 (35) |
| 51 to 59 years | 59 (30.1) | 51 (27.2) | 110 (28.7) |
| ≥60 years | 64 (32.7) | 56 (30.0) | 120 (31.3) |
| Unknown | 10 (5.1) | 9 (4.8) | 19 (5.0) |
| **Kinship, $n$ (%)** | | | |
| Parents | 10 (5.1) | 12 (6.4) | 22 (5.7) |
| Offspring | 17 (8.7) | 30 (7.8) | 13 (7.0) |
| Siblings | 169 (86.2) | 162 (86.6) | 331 (86.5) |
| **Inclusion criteria, $n$ (%)** | | | |
| One index case <60 years | 147(75.0) | 141(75.4) | 288 (75.2) |
| Two or more index cases | 14 (7.1) | 20 (10.7) | 34 (8.9) |
| Siblings | 103 (52.5) | 113 (60.4 | 216 (56.4) |
| **Comorbidity[c], $n$ (%)** | | | |
| No | 178 (90.8) | 166 (88.8) | 344 (89.8) |
| Yes | 21 (11.2 | 18 (9.2) | 39 (10.2) |
| **Smoker status, $n$ (%)** | | | |
| Never smoke | 160 (81.6) | 151 (80.7) | 311 (81.2) |
| Ever smoker | 31 (15.8) | 32 (17.1) | 63 (16.5) |
| Unknown | 5 (2.1) | 4(2.6) | 9 (2.3) |
| **Alcohol consumption, $n$ (%)** | | | |
| No | 140 (71.4) | 147 (78.6) | 287 (75.0) |
| Yes | 42 (21.4) | 32 (17.1) | 74 (19.3) |
| Unknown | 14 (7.1) | 8 (4.3) | 22 (5.7) |
| **Aspirin use, $n$ (%)** | 10 (5.2) | 4 (2.1) | 14 (3.7) |
| **Anticoagulants use, $n$ (%)** | 3 (2.0) | 2 (1.2) | 5 (1.6) |
| **Educational level, $n$ (%)** | | | |
| No studies | 25 (12.8) | 24 (12.8) | 49 (12.8) |
| Primary level | 82 (41.8) | 72 (38.5) | 154 (40.2) |
| Secondary level | 53 (27.0) | 51 (27.3) | 104 (27.2) |
| University level | 17 (8.7) | 23 (12.3) | 40 (10.4) |
| Unknown | 19 (9.7) | 17 (9.1) | 36 (9.4) |
| **Index cases** | $N = 460$ | | |
| Mean age at CRC[d] diagnosis ± SD | 57.6 ± 9.6 | | |
| Male, $n$ (%) | 269 (58.4) | | |
| Female, $n$ (%) | 191 (41.6) | | |
| CRC site, $n$ (%) | | | |
| Rectum | 121 (26.3) | | |

*(Continued)*

**Table 1.** (Continued)

| Category | Colonoscopy group | FIT[b] group | Total |
|---|---|---|---|
| Colon | 339 (73.7) | | |

Data of participants were stratified according to as-screened analysis.

[a] FDR = first-degree relatives.

[b] FIT = fecal immunochemical test.

[c] Comorbidity was considered if there was at least any chronic disease (diabetes mellitus, hypertension, cardiopathy, chronic renal disease, pulmonary disease, or neoplastic disease). Alcohol consumption was considered if 1 or more drinks per week were registered.

[d] CRC = colorectal cancer.

The total numbers in some categories might exceed the total number of subjects, because FDR might have more than 1 relative with CRC, or had the double condition of having an index case younger than 60 and being siblings. When 2 index cases were identified in the same family, the younger patient was considered the reference index case for the study.

colonoscopy work-up if a positive test (Fig 1). Among the 431 subjects assigned to colonoscopy, 196 (45.5%) agreed to participate, and 147 (34.1%) of them underwent colonoscopy (Fig 1). The demographic characteristics of FDR that underwent screening compared with those who refused to participate after signing the informed consent are shown in Table 3. Among the 383 FDR who signed the informed consent there were 78 (20.4%) that refused to

**Table 2. Demographic data of FDR[a] noncompliant with the assigned strategy.**

| Category | Colonoscopy group | FIT[b] group | Total | | | |
|---|---|---|---|---|---|---|
| FDR | N = 49 | N = 29 | N = 78 | Odds ratio | 95% CI[c] | p-value |
| Mean age ± SD | 57.5 ± 10.3 | 55.0 ± 10.9 | 56.6 ± 10.5 | | (−3.31, 6.55) | 0.516 |
| Male, n (%) | 22 (44.9) | 16 (55.2) | 38 (48.7) | 0.66 | (0.26, 1.67) | 0.380 |
| Comorbidity[d], n (%) | | | | | | |
| No | 46 (93.9) | 29 (100) | 75 (96.2) | 1.06 | (0.99, 1.14) | 0.174 |
| Yes | 3 (6.1) | 0 (0) | 3 (3.8) | | | |
| Smoker status, n (%) | | | | | | |
| Never smoke | 40 (81.6) | 25 (86.2) | 65 (83.3) | 1.98 | (0.35, 10.03) | 0.462 |
| Ever smoker | 6 (12.2) | 2 (6.9) | 8 (10.3) | | | |
| Unknown | 3 (6.1) | 2 (6.9) | 5 (6.4) | | | |
| Alcohol consumption, n (%) | | | | | | |
| No | 39 (79.6) | 23 (79.3) | 62 (79.5) | 0.59 | (0.11, 3.16) | 0.538 |
| Yes | 3 (6.1) | 3 (10.3) | 6 (7.7) | | | |
| Unknown | 7 (14.3) | 3 (10.3) | 10 (12.8) | | | |
| Educational level, n (%) | | | | | | |
| No studies | 5 (10.2) | 0 (0.0) | 5 (6.4) | 1.14 | (1.01, 1.29) | 0.096 |
| Primary level | 20 (40.8) | 10 (34.5) | 30 (38.5) | 1.05 | (0.35, 3.09) | 0.926 |
| Secondary level | 13 (26.6) | 7 (24.1) | 20 (25.6) | 0.93 | (0.29, 2.88) | 0.898 |
| University level | 1 (2.0) | 3 (10.4) | 4 (5.1) | 0.15 | (0.01, 1.54) | 0.110 |
| Unknown | 10 (20.4) | 9 (31.0) | 19 (24.4) | 0.57 | (0.19, 1.62) | 0.294 |

Data were assessed according to as-screened analysis.

[a] FDR = first-degree relatives.

[b] FIT = fecal immunochemical test.

[c] CI = confidence interval.

[d] Comorbidity was considered if there was at least any chronic disease (diabetes mellitus, hypertension, cardiopathy, chronic renal disease, pulmonary disease, or neoplastic disease). Alcohol consumption was considered if one or more drinks per week were registered.

**Table 3. Demographic characteristics of subjects that underwent screening compared to those that refused screening after signing the informed consent (unadjusted analysis).**

| Categories | | Subjects that underwent screening N = 305 | Subjects that refused screening N = 78 | Odds ratio | 95% CI[a] | p-value |
|---|---|---|---|---|---|---|
| Mean Age ± SD | | 55.0 ± 9.9 | 56.6 ± 10.53 | -- | - 0.91, 4.12 | 0.211 |
| Sex, n (%) | Femal | 146 (48.9) | 40 51.3 | 1.14 | 0.69, 1.88 | 0.591 |
| | Male | 159 (52.1) | 38 (48.7) | | | |
| Participating centers [b], n (%) | High recruitment | 279 (91.5) | 77 (98.7) | 0.13 | 0.01, 1.04 | 0.026 |
| | Low recruitment | 26 (8.5) | 1 (1.3) | | | |
| Smoker status, n (%) | Ever smoker | 55 (18) | 8 (10.3) | 1.81 | 0.82, 4.00 | 0.134 |
| | Non-smoker | 246 (80.7) | 65 (83.3) | | | |
| | Unknown | 4 (1.3) | 5 (6.4) | | | |
| Alcohol consumption[c], n (%) | Yes | 68 (22.3) | 6 (7.7) | 3.12 | 1.29, 7.53 | 0.008 |
| | No | 225 (73,7) | 62 (79.5) | | | |
| | Unknown | 12 (4) | 10 (12.8) | | | |
| Comorbidity[d], n (%) | Yes | 36 (11.8) | 3 (3.8) | 3.34 | 1.00, 11.16 | 0.050 |
| | No | 269 (88.2) | 75 (96.2) | | | |
| **Kinship, n (%)** | | | | | | |
| Parents | Yes | 17 (5.5) | 5 (6.4) | 0.86 | 0.30, 2.41 | 0.777 |
| | No | 288 (94.5) | 73 (93.6) | | | |
| Offspring | Yes | 27 (8.9) | 3 (3.8) | 2.42 | 0.71, 8.22 | 0.142 |
| | No | 278 (91.1) | 75 (96.2) | | | |
| Siblings | Yes | 261 (85.6) | 70 (89.7) | 0.67 | 0.30, 1.50 | 0.337 |
| | No | 44 (14.4) | 8 (10.3) | | | |
| **Educational level n(%)** | | | | | | |
| No studies or Primary level | Yes | 168 (55) | 35 (44.9) | 0.96 | 0.54, 1.69 | 0.888 |
| | No | 120 (39.4) | 24 (30.8) | | | |
| | Unknown | 17 (5.6) | 19 (24.3) | | | |
| Secondary level | Yes | 84 (27.6) | 20 (25.7) | 0.80 | 0.44, 1.45 | 0.470 |
| | No | 204 (66.8) | 39 (50) | | | |
| | Unknown | 17 (5.6) | 19 (24.3 | | | |
| University level | Yes | 36 (11.8) | 4 (5.2) | 1.96 | 0.67, 5.74 | 0.210 |
| | No | 252 (82.6) | 55 (70.5) | | | |
| | Unknown | 17 (5.6) | 19 (24.3) | | | |

[a] CI = confidence interval.

[b] Participating centers were categorized as high or low recruiters, if they included more or less than 80 eligible FDR in the study, respectively.

[c] Alcohol consumption was considered if one or more drinks per week were registered.

[d] Comorbidity was considered if there was at least any chronic disease (diabetes mellitus, hypertension, cardiopathy, chronic renal disease, pulmonary disease, or neoplastic disease).

participate. Both cohorts were similar regarding age, sex, smoker status, kinship, and educational level. FDR compliant with the assigned strategy had a higher rate of alcohol consumption and more comorbidity than those who refused screening.

The uptake of colonoscopy screening (34.1%) was similar to the uptake of FIT screening (35.9%) in the bivariate analysis (OR = 1.08, 95% CI [0.82, 1.43], $p$ = 0.560). In the multiple logistic regression analysis, the screening strategy, FDR's sex and age, the location of the tumor in the index case, the person who attended the first appointment to participate in the trial (index case or other FDR), and degree of recruitment of participating centers were not significantly associated with screening uptake. The adjusted analysis showed that assignment of a

**Table 4. Odds of screening uptake[a] during the recruitment period (2015–2019), according to intention-to-screen analysis.**

| | | Eligible FDR[b] (N = 870) | Screening Uptake N (%) | Unadjusted analysis | | Adjusted[c] analysis | |
|---|---|---|---|---|---|---|---|
| | | | | Odds ratio (95% CI[d]) | p-value | Odds ratio (95% CI) | p-value |
| **Screening strategy** | Colonoscopy | 431 | 147 (34.1) | 1.08 (0.82, 1.43) | 0.560 | 1.08 (0.82, 1.44) | 0.564 |
| | FIT[d] | 439 | 158 (35.9) | | | | |
| **FDR, mean age ± SD** | 55.0 ± 9.9 | 305 | 305 (35.0) | 1.01 (1.00, 1.03) | 0.123 | 1.01 (1.00, 1.03) | 0.104 |
| | 56.1 ± 10.7 | 565 | 565 (65.0) | | | | |
| **FDR, sex** | Male | 419 | 159 (37.9) | 1.27 (0.96, 1.68) | 0.085 | 1.29 (0.98, 1.72) | 0.078 |
| | Female | 451 | 146 (32.4) | | | | |
| **Same vs different strategies assigned per family** | Same strategy | 266 | 114 (42.9) | 1.63 (1.20, 2.22) | 0.001 | 1.66 (1.23, 2.56) | 0.001 |
| | Different strategies | 604 | 191 (31.6) | | | | |
| **Index case, tumor location** | Colon | 650 | 233 (35.8) | 1.14 (0.83, 1.58) | 0.402 | 1.10 (0.79, 1.54) | 0.539 |
| | Rectum | 220 | 72 (32.7) | | | | |
| **Person who attended the first appointment** | Index case | 783 | 205 (35.1) | 1.03 (0.64, 1.56) | 0.906 | 1.06 (0.66, 1.71) | 0.797 |
| | Other FDR | 87 | 30 (34.5) | | | | |
| **Participating centers[f]** | High recruitment | 787 | 279 (35.5) | 1.20 (0.74, 1.95) | 0.454 | 0.80 (0.48, 1.31) | 0.383 |
| | Low recruitment | 83 | 26 (31.3) | | | | |

[a] Screening uptake was considered when a subject underwent at least one FIT and colonoscopy work-up, in case of a positive test in the FIT, group and if it was compliant with one-time colonoscopy in the colonoscopy group, along the recruitment period.

[b] FDR = first-degree relative.

[c] The multiple logistic regression analysis was adjusted by screening strategy, sex and age of FDR, assignment of the same or different screening strategies per family, index case tumor location, person who attended the first appointment and degree of recruitment of participating centers.

[d] CI = confidence interval.

[e] FIT = fecal immunochemical test.

[f] Participating centers were categorized as high or low recruiters, if they included more or less than 80 eligible FDR in the study, respectively.

different strategy (FIT or colonoscopy) in FDRs from the same family negatively influenced the overall participation in the study (OR = 1.66, 95% CI [1.23, 2.56], p = 0.001) (Table 4). In this regard, the rate of subjects who were assigned to different strategies in the same family was similar in the FIT group (310/604, 51.3%) and in the colonoscopy group (294/604, 48.7%) (OR = 0.89, 95% CI [0.66, 1.19], p = 0.442]. In addition, screening uptake did not differ between arms, in the subgroup of FDR that were assigned the same/one strategy (OR = 0.95, 95% CI [0.58, 1.55], p = 0.859) or in those that were assigned different strategies (OR = 0.88, 95% CI [0.62, 1.24], p = 0.487) in the family (S2 Table).

The detection rate of advanced colorectal neoplasia was 5.6% (n = 24) in the colonoscopy group and 1.6% (n = 7) in the FIT group (OR 3.64, 95% CI [1.55, 8.53], p = 0.003) (Table 5).

After adjusting for potential confounders (age, sex, and center), we found that the colonoscopy group had significantly increased odds of advanced colorectal neoplasia compared with the FIT group (OR 3.53; 95% CI [1.49, 8.32], p = 0.004) (Table 6).

As of data cutoff (December 31, 2021), the median follow-up was 46.4 months (IQR 36.4 to 54.9), 49.9 months (IQR 39.8 to 58.2) in the colonoscopy group, and 40.0 months (IQR 34.4 to 50.8) in the FIT group.

Table 7 shows the follow-up data of FDR that agreed to participate. Overall, 305/383 (79.6%) subjects complied with the assigned strategy. During follow-up, 111/305 (36.4%) kept screening with the assigned method with no significant differences between the 2 groups.

**Table 5. Detection rate of colorectal neoplasia according to intention-to-screen analysis.**

| Findings | Colonoscopy group (N = 431) | FIT[a] group (N = 439) | Odds ratio | 95% CI[b] | p-value |
|---|---|---|---|---|---|
| Non-advanced adenomas, n (%) | 32 (7.4) | 9 (2.0) | 3.83 | (1.80, 8.12) | 0.001 |
| Non-advanced sessile serrated lesions[c], n (%) | 4 (0.9) | 0 (0.0) | 1.0 | (1.00, 1.02) | 0.043 |
| Advanced adenoma, n (%) | 22 (5.1) | 6 (1.4) | 3.9 | (1.56, 9.67) | 0.004 |
| Invasive CRC[d], n (%) | 2 (0.5) | 1 (0.2) | 2.0 | (0.18, 22.60) | 0.561 |
| Advanced colorectal neoplasia[e], n (%) | 24 (5.6) | 7 (1.6) | 3.64 | (1.55, 8.53) | 0.003 |

In the intention-to-screen analysis, the detection rate of neoplastic lesions was calculated as the number of subjects with true positive results divided by the number of eligible FDR. Subjects were classified according to the most advanced lesion.

[a] FIT = fecal immunochemical test.

[b] CI = confidence interval.

[c] Non-advanced sessile serrated lesions = polyps without dysplasia and <10 mm in size.

[d] CRC = colorectal cancer.

[e]Advanced colorectal neoplasia comprised 22 advanced adenomas (measuring 10 mm or more in diameter), and 2 invasive or CRC.

Considering the FDR that did not comply with the assigned strategy in the trial, but that were screened during follow-up (unplanned FIT or colonoscopy screening), either at the request of their general practitioner or at their own request, a total of 166 individuals received screening with colonoscopy (38.5%) and 180 received screening with FIT (41.0%).

The multivariable logistic regression model showed that the screening strategy, sex, age, and participating centers had no effect on the screening uptake at the end of the follow-up period (Table 8). In addition, the cumulative detection rate of advanced colorectal neoplasia at the end of follow-up was significantly higher in subjects assigned to colonoscopy screening versus those undergoing FIT screening (Table 9). No major complications were associated with colonoscopies performed during the procedure or in the following 30 days. There were 5

**Table 6. Odds of advanced colorectal neoplasia[a] during the recruitment period (2015–2019) according to intention-to-screen analysis.**

| | | Eligible FDR[c] (N = 870) | Advanced colorectal neoplasia N (%) | Unadjusted analysis | | Adjusted[b] analysis | |
|---|---|---|---|---|---|---|---|
| | | | | Odds ratio (95% CI[d]) | p-value | Odds ratio (95% CI) | p-value |
| Screening strategy | Colonoscopy | 431 | 24 (5.6) | 3.64 (1.55, 8.54) | 0.003 | 3.53 (1.49, 8.32) | 0.004 |
| | FIT[e] | 439 | 7 (1.6) | | | | |
| Sex | Male | 419 | 21 (5.0) | 2.32 (1.08, 5.00) | 0.026 | 2.28 (1.05, 4.95) | 0.037 |
| | Female | 451 | 10 (2.2) | | | | |
| Age[f] | ≥55 years | 430 | 23 (5.3) | 3.05 (1.35, 6.9) | 0.007 | 2.81 (1.23, 6.41) | 0.014 |
| | <55 years | 440 | 8 (1.8) | | | | |
| Participating center[g] | High recruitment | 787 | 29 (3.7) | 1.54 (0.36, 6.619) | 0.551 | 1.84 (0.42, 8.00) | 0.416 |
| | Low recruitment | 83 | 2 (2.4) | | | | |

[a] The diagnostic yield of advanced colorectal neoplasia was the number of subjects with true positive results (advanced adenoma or serrated polyp and/or CRC) divided by the number of eligible subjects according to the intention-to-screen analysis.

[b] The multiple logistic regression analysis was adjusted by screening strategy, sex, age, and degree of recruitment of participating centers.

[c] FDR = first-degree relatives.

[d] CI = confidence interval.

[e] FIT = fecal immunochemical test.

[f] Age was categorized according to the median age of the eligible population.

[g] Participating centers were categorized as high or low recruiters, if they included more or less than 80 eligible FDR in the study, respectively.

**Table 7. Follow-up data of FDR[a] that agreed to participate in each study group.**

| Categories | Colonoscopy group (N = 196) | | FIT[b] group (N = 187) | | Total (N = 383) | | Odds ratio | 95% CI[c] | p-value |
|---|---|---|---|---|---|---|---|---|---|
| | Subjects, n | Rate, (%) | Subjects, n | Rate, (%) | Subject, n | Rate, (%) | | | |
| **Compliant[d]** | **147** | **100** | **158** | **100** | **305** | **100** | **0.55** | **(0.33, 0.91)** | **0.210** |
| Continued on assigned strategy | 54 | 36.7 | 57 | 36 | 111 | 36.4 | 1.03 | (0.64, 1.64) | 0.905 |
| Unplanned colonoscopy screening[e] | 2 | 1.4 | 20 | 12.6 | 22 | 7.2 | 0.15 | (0.04, 0.52) | 0.001 |
| Unplanned FIT screening[f] | 20 | 13.6 | 9 | 5.7 | 29 | 9.5 | 2.33 | (1.05, 5.16) | 0.033 |
| Interval colonoscopies[g] | 3 | 2.0 | 6 | 3.8 | 9 | 2.9 | 0.34 | (0.07, 1.76) | 0.183 |
| Postpolypectomy surveillance | 21 | 14.3 | 3 | 1.9 | 24 | 7.9 | 8.61 | (2.51, 29.53) | 0.001 |
| Lost to follow-up[h] | 42 | 28.6 | 62 | 39.2 | 104 | 34.1 | 0.61 | (0.38, 100) | 0.050 |
| Deceased | 5 | 3.4 | 1 | 0.6 | 6 | 2 | 5.52 | (0.63, 47.88) | 0.082 |
| **Noncompliant[i]** | **49** | **100** | **29** | **100** | **78** | **100** | **0.55** | **(0.33, 0.91)** | **0.210** |
| Unplanned colonoscopy screening | 13 | 26.5 | 6 | 20.7 | 19 | 24.4 | 1.38 | (0.46, 4.15) | 0.561 |
| Unplanned FIT screening | 17 | 34.7 | 5 | 17.2 | 22 | 28.2 | 2.55 | (0.82, 7.88) | 0.098 |
| Interval colonoscopies | 1 | 2.0 | 0 | 0.0 | 1 | 1.2 | 1.02 | (0.98, 1.06) | 0.439 |
| Lost to follow-up | 18 | 36.7 | 18 | 62 | 36 | 46.2 | 0.35 | (0.13, 0.91) | 0.030 |

[a] FDR = first-degree relatives.

[b] FIT = fecal immunochemical test.

[c] CI = confidence interval.

[d] Compliant = individuals that signed the informed consent and completed the assigned strategy.

[e] Unplanned colonoscopy screening = individuals that received unplanned colonoscopy screening in either study group.

[f] Unplanned FIT = any FIT performed as a screening tool group outside the trial design.

[g] In the FIT group, an interval colonoscopy was defined as any colonoscopy performed after a negative FIT result. In the colonoscopy group, interval colonoscopy referred to colonoscopies performed within 36 months after a baseline colonoscopy.

[h] Lost to follow-up were considered FDR compliant or not compliant during the inclusion period (February 2016 to December 2019), which had not additional information in their data records during the follow-up period (January 2020 to December 2021.

[i] Noncompliant = individuals that signed the informed consent but later declined the assigned testing.

deceased FDR in the colonoscopy group and 1 in the FIT group (OR 5.52, 95% CI [0.63, 47.88], p = 0.083) (Table 7). Three of them, belonging to the colonoscopy group were detected during the follow-up period but were subjects that did not participate in the trial. So, there were no interval cancers throughout the study.

The detection rate of lesions in the FIT group following completion of the first, second, or third tests according to the as-screened analysis is shown in Table 10. Of the 23 FDR with a positive FIT, 17 (73·9%), 5 (21·7%), and 1 (4·3%) were found in the first, second, and third round, respectively. Overall, 19/23 (82·6%) subjects with a positive FIT underwent colonoscopy, showing 6 (31·6%) advanced adenomas and 1 (5·3%) CCR. No major complications were associated with colonoscopies performed during the procedure or in the following 30 days.

## Discussion

In this multicenter randomized controlled trial, screening uptake of FIT was not different than screening uptake of colonoscopy among FDR with a high-risk family history of non-syndromic CRC. In addition, the detection rate of advanced colorectal neoplasia was significantly higher among subjects undergoing screening colonoscopy than in those receiving FIT screening. Therefore, the results did not support the hypothesis that FIT screening might be better

**Table 8. Odds of overall screening uptake[a] along the recruitment and the follow-up periods (2015–2021) according to intention-to-screen analysis.**

| | | Eligible FDR[c] (N = 870) | Screening Uptake N (%) | Unadjusted analysis Odds ratio (95% CI[d]) | p-value | Adjusted[b] analysis Odds ratio (95% CI) | p-value |
|---|---|---|---|---|---|---|---|
| **Screening strategy** | Colonoscopy | 431 | 177 (41.1) | 1.11 (0.84, 1.86) | 0.439 | 1.12 (0.84, 1.47) | 0.440 |
| | FIT[e] | 439 | 169 (38.5) | | | | |
| **FDR, mean age ± SD** | 55.0 ± 9.9 | 305 | 305 (35.0) | 1.01 (1.00, 1.03) | 0.242 | 1.01 (1.00, 1.03) | 0.212 |
| | 56.1 ± 10.7 | 565 | 565 (65.0) | | | | |
| **FDR, sex** | Men | 419 | 177 (42.2) | 1.23 (0.93, 1.61) | 0.151 | 1.23 (0.94, 1.64) | 0.137 |
| | Female | 451 | 169 (37.5) | | | | |
| **Same vs Different strategies per family** | Same strategy | 266 | 127 (47.7) | 1.61 (1.16, 2.32) | 0.001 | 1.66 (1.23, 2.23) | 0.001 |
| | Different strategies | 604 | 219 (36.3) | | | | |
| **Index case, tumor location** | Colon | 650 | 261 (40.2) | 1.07 (0.78, 1.47) | 0.691 | 1.03 (0.75, 1.42) | 0.839 |
| | Rectum | 220 | 85 (38.6) | | | | |
| **Person who attended the first appointment** | Index case | 783 | 312 (39.8) | 1.04 (0.65, 1.63) | 0.890 | 1.05 (0.66, 1.67) | 0.833 |
| | Other FDR | 87 | 34 (39.1) | | | | |
| **Participating centers[f]** | High recruitment | 787 | 320 (40.7) | 1.50 (0.92, 2.44) | 0.098 | 1.58 (0.97, 2.63) | 0.07 |
| | Low recruitment | 83 | 26 (31.3) | | | | |

[a] Overall screening uptake was considered when a subject underwent at least one FIT, and colonoscopy work-up in case of a positive test in the FIT group, and if it was compliant with one-time colonoscopy in the colonoscopy group, along the recruitment and the follow-up periods.

[b] The multiple logistic regression analysis was adjusted by screening strategy, sex and age of FDR, assignment of the same or different screening strategies per family, index case tumor location, person who attended the first appointment, and degree of recruitment of participating centers.

[c] FDR = first-degree relatives.

[d] CI = confidence interval.

[e] FIT = fecal immunochemical test.

[f] Participating centers were categorized as high or low recruiters, if they included more or less than 80 eligible FDR in the study, respectively

**Table 9. Odds of overall advanced colorectal neoplasia[a] along the recruitment and the follow-up periods (2015–2021) according to intention-to-screen analysis.**

| | | Eligible FDR[c] (N = 870) | Advanced colorectal neoplasia N (%) | Unadjusted analysis Odds ratio (95% CI[d]) | p-value | Adjusted[b] analysis Odds ratio (95% CI) | p-value |
|---|---|---|---|---|---|---|---|
| Screening strategy | Colonoscopy | 431 | 27 (6.3) | 2.59 (1.27, 5.31) | 0.007 | 2.50 (1.21, 5.15) | 0.013 |
| | FIT[e] | 439 | 11 (2.5) | | | | |
| Sex | Male | 419 | 25 (6.0) | 2.13 (1.07, 4.23) | 0.026 | 2.08 (1.04, 4.16) | 0.037 |
| | Female | 451 | 13 (2.9) | | | | |
| Age[f] | ≥55 years | 430 | 29 (6.7) | 3.46 (1.62, 7.40) | 0.001 | 3.28 (1.52, 7.09) | 0.002 |
| | <55 years | 440 | 9 (2.0) | | | | |
| Participating centers[g] | High recruitment | 787 | 36 (4.6) | 1.94 (0.45, 8.21) | 0.359 | 1.46 (0.54, 10.1) | 0.253 |
| | Low recruitment | 83 | 2 (2.4) | | | | |

[a] The overall detection rate of advanced colorectal neoplasia was defined as the number of subjects with true positive results (advanced adenoma and/or CRC) divided by the number of eligible subjects, according to the intention-to-screen analysis, along the recruitment and the follow-up periods.

[b] The multiple logistic regression analysis was adjusted by sex, age, and degree of recruitment at participating centers.

[c] FDR = first-degree relatives.

[d] CI = confidence interval.

[e] FIT = fecal immunochemical test.

[f] Age was categorized according to the median age of the eligible population.

[g] Participating centers were categorized as high or low recruiters, if they included more or less than 80 eligible FDR in the study, respectively.

**Table 10. Diagnostic yield in FDR[a] with a positive FIT[b] following completion of the first, second, or third tests according to as-screened analysis.**

| Variable | First FIT ($n = 162$) | Second FIT ($n = 84$) | Third FIT ($n = 67$) |
|---|---|---|---|
| Positive FIT result, $n$ (%) | 17 (10·5) | 5 (6.0) | 1 (1.5) |
| Complete colonoscopy[c], $n$ (%) | 15 (9.3) | 3 (3.6) | 1 (1.5) |
| **Colonoscopy result, $n$ (%):** | | | |
| Normal or non-neoplastic lesions | 1 (0.6) | 1 (1.2) | 1 (1.5) |
| Non-advanced adenomas | 9 (5.5) | - | - |
| Advanced adenomas[d] | 5 (3.0) | 1 (1.2) | - |
| - Invasive CRC[e] | - | 1 (1.2) | - |
| Advanced colorectal neoplasia[f] | 5 (3.0) | 1 (1.2) | - |

[a] FDR = first-degree relatives.

[b] FIT = fecal immunochemical test.

[c] Complete colonoscopy = colonoscopy that reached the cecum and had adequate bowel preparation (at least 90% of the mucosal surface was explored).

[d] Advanced adenoma = adenoma measuring ≥10 mm in diameter, with tubulovillous architecture, high-grade dysplasia, or intramucosal carcinoma.

[e] CRC = colorectal cancer.

[f] Advanced colorectal neoplasia = adenoma or serrated polyps measuring 10 mm or more in diameter, with tubulovillous architecture, high-grade dysplasia, in situ adenocarcinoma and/or CRC.

accepted and equally effective as colonoscopy screening for detecting advanced colorectal neoplasia in this population.

Our study has several strengths. First, randomization was performed before the initial appointment to avoid selection bias. Second, we included only asymptomatic FDR from index cases with non-syndromic CRC diagnosed no more than 2 years before the start of the study. This assured that they were within the frame of the recommendations of current guidelines. Third, family history was verified through the index case, and only FDR not previously screened were included. Therefore, using the index case as the provider of the family history assured that all eligible relatives could be contacted and helped to avoid the pitfalls of self-reported enrollment. Fourth, we included eligible relatives from 7 Spanish Autonomous regions, suggesting that the results might be extrapolated to familial CRC population in Spain. Fifth, colonoscopies and bowel cleansing preparation were offered free of charge in both study groups, which could facilitate participation in the study. Sixth, extended follow-up until December 2021 allowed us to identify participants that were screened outside the trial context, minimizing the negative effect that COVID-19 pandemic had on colon cancer screening during 2020.

On the other hand, the study also has some limitations. First, 536/870 (56.0%) eligible FDR (57.4% in the FIT group and 54.5% in the colonoscopy group) declined to participate and did not attend the initial appointment. We could not contact these individuals as Spanish law does not allow registering of data of individuals that have not given previous informed consent. Although a reminder letter was mailed to the index case to encourage the participation of their non-attending relatives, we cannot rule out delivery failure in some cases. Second, among the 383 FDR who signed the informed consent form, 78 (20.3%) refused to undergo screening. However, follow-up information from these individuals allowed us to estimate the rate of screening outside the trial protocol, which provided a more accurate global uptake in both study groups according to intention-to-screen analysis. Third, randomization was performed on the eligible subjects and not by family cluster to guarantee a homogeneous sample in each

group of the study. Consequently, some individuals had a different screening strategy assigned in the same family, which could have led to refusal to participate in some cases. In fact, the assignment of different strategies to members of the same family was an independent factor for low participation in the logistic regression analysis. However, the rate of subjects who were assigned to different strategies in the same family was similar in the FIT group (51.3%) and in the colonoscopy group (48.7%). In addition, the screening uptake did not differ between the study groups that were assigned the same strategy or different strategies in the family (S2 Table), suggesting that this condition affected both groups equally.

Annual or biennial FIT is the most widely used screening strategy in countries and health organizations with organized CRC screening programs [26,27]. Recently, it has been suggested that repeated FIT could be an alternative to colonoscopy screening in the familial-risk population, which could overcome the suboptimal uptake of colonoscopy in these individuals. This hypothesis has been formulated under the following premises. First, large prospective cohort studies have shown that only subjects with 2 or more FDR affected in the immediate family had a significantly higher risk of advanced colorectal neoplasia, compared to average risk individuals [3,28]. In addition, a pooled analysis of 6 prospective cohort studies showed that family history of CRC is not associated with overall survival or CRC-specific survival after adjusting for confounders [4]. Overall, these studies suggest that most FDR of patients with CRC would not benefit from intensive colonoscopy surveillance and could be screened in the same way as the average-risk population. Second, FIT screening may be equivalent to colonoscopy screening for detecting CRC and advanced adenomas in familial-risk population. This is supported by a meta-analysis that included 11 observational studies and a randomized controlled trial [14]. Third, assuming that FIT is a reasonably well-accepted screening procedure in the average-risk population, it has been proposed that it could also be extended to the familial-risk population. However, this hypothesis has not been evaluated in a clinical trial.

The current study shows that in Spain, a country with universal public health care and more than a decade of experience with a FIT-based nationwide screening program, FIT screening did not improve the screening acceptance compared to colonoscopy screening in the familial-risk population. The low uptake of colonoscopy screening in our study was to be expected if we compare it with those from European studies performed in the average-risk or in the familial-risk population [10,29,30]. However, we did not expect such low acceptance of FIT screening. Adherence was already low (35.9%) in the first screening round but dropped to 17% for individuals that completed the 3 tests, which is unacceptable for any FIT screening program. Although we could not obtain information of the subjects who did not attend the invitation to participate in the study, we had data of 78/383 (20.4%) individuals that refused to participate after signing the informed consent. The comparison of demographic data between participants and this representative sample of non-participants was similar regarding age, sex, smoker status, kinship, and educational level. Therefore, these conditions do not seem to clarify the reasons why these subjects refused screening.

A relevant aspect that could at least partially explain the low uptake rate observed in our study is that the invitation was formulated in an opportunistic setting. However, this is not different from what occurs in real clinical practice. Traditionally, FDR of patients with CRC have been excluded from nationwide screening programs because they are considered candidates for straightforward colonoscopy. Paradoxically, this approach is associated with a suboptimal acceptance rate, leaving a substantial number of these individuals unscreened. Nevertheless, the reasons why more than half of subjects with a high-risk family history of CRC refused to be screened are unknown and cannot be ascertained by our study for the reasons mentioned above.

Among the major barriers for screening adherence or for non-follow-up with colonoscopy after a positive FIT in these individuals could be the lack of symptoms, low knowledge of one's risk for developing CRC, decision-making difficulties, and low provider awareness about recommendations established by clinical guidelines [31]. One step forward to improve the screening uptake of these individuals could be to involve them in organized screening programs. In fact, a study performed in the setting of a Dutch screening program revealed that providing familial risk assessment to individuals with a positive FIT may facilitate the identification of high-risk FDR and prevent the development of a substantial number of CRC cases [32]. However, this approach would not improve the participation of those who decline to be screened or have a negative FIT. In line with this finding, a recent meta-analysis of 4 controlled trials [33] showed that tailored communication based on written and verbal information increased the participation rate in colonoscopy screening by about 2-fold. In addition, a recent trial performed within the framework of the Polish Colonoscopy Screening Program have shown that offering screening strategies that combine FIT and colonoscopy can result in participation rates 8% to 10% points higher compared to offering colonoscopy screening alone [34]. Despite the design of our study did not allow for changing the randomly assigned group, 34% of subjects that were noncompliant with colonoscopy screening and 20.7% of those noncompliant with FIT screening crossed over to the other group during the follow-up period. This suggests that the screening uptake of this population could have improved if both options had been offered together.

As expected, the similar screening uptake of straightforward colonoscopy and annual FIT screening observed in our study was associated with a significantly higher detection rate of advanced neoplasia in individuals assigned to one-time colonoscopy compared to those screened by FIT. This finding differs from previous studies suggesting that FIT screening might be equivalent to colonoscopy screening for detecting advanced neoplasia in this population [15]. This discrepancy can be explained by the fact that in previous studies, recruitment was carried out among family members who were willing to be screened, while in the current study, all eligible family members were included in a more pragmatic intention-to-screen analysis. In addition, the current study was performed in FDR with a high-risk family history of CRC, whereas previous studies included most relatives at low or moderate risk, which could justify different detection rates of advanced colorectal neoplasia. Nevertheless, these data should be analyzed with caution as only a very low number (29 individuals) fulfilled 3 round of testing and only 19 colonoscopies were performed in the FIT group. So, detection rates of advanced colorectal neoplasia might be based on chance in these individuals.

In conclusion, this randomized controlled trial indicates that in the setting of an opportunistic screening, annual FIT does not increase the screening uptake compared to colonoscopy screening in FDR at high risk of developing CRC, resulting in a significantly lower detection rate of advanced colorectal neoplasia. New initiatives are needed to assess whether screening uptake can be improved for these individuals through their inclusion in population-based screening programs or by offering a choice between fit and colonoscopy screening.

## Supporting information

**S1 CONSORT Checklist. Consolidated Standards of Reporting Trials.**
(DOC)

**S1 Text. Study protocol.**
(DOCX)

**S2 Text. Ethics Committee Approval Letter.**
(PDF)

**S3 Text. Study protocol (original version).**
(DOCX)

**S1 Table. Computation of futility analysis.**
(DOCX)

**S2 Table. Eligible first-degree relatives randomly assigned to the same screening strategy or to different strategies in the family.**
(DOCX)

**S1 Data. ParCoFit study database.**
(CSV)

## Acknowledgments

We thank all the staff of the endoscopic units from the 12 participating centers for their efforts in collecting and preparing the data for this study.

## Author Contributions

**Conceptualization:** Enrique Quintero, Antonio Z. Gimeno-Garcia.

**Data curation:** Natalia González-López.

**Formal analysis:** Natalia González-López, Enrique Quintero, Enrique González-Dávila.

**Funding acquisition:** Antonio Z. Gimeno-Garcia.

**Investigation:** Natalia González-López, Enrique Quintero, Antonio Z. Gimeno-Garcia, Luis Bujanda, Jesús Banales, Joaquin Cubiella, María Salve-Bouzo, Jesus Miguel Herrero-Rivas, Estela Cid-Delgado, Victoria Alvarez-Sanchez, Alejandro Ledo-Rodríguez, Maria Luisa de-Castro-Parga, Romina Fernández-Poceiro, Luciano Sanromán-Álvarez, Jose Santiago-Garcia, Alberto Herreros-de-Tejada, Teresa Ocaña-Bombardo, Francesc Balaguer, María Rodríguez-Soler, Rodrigo Jover, Marta Ponce, Cristina Alvarez-Urturi, Xavier Bessa, Maria-Pilar Roncales, Federico Sopeña, Angel Lanas, David Nicolás-Pérez, Zaida Adrián-de-Ganzo, Marta Carrillo-Palau.

**Methodology:** Enrique Quintero.

**Resources:** Enrique Quintero.

**Software:** David Nicolás-Pérez.

**Supervision:** Enrique Quintero, Luis Bujanda, Joaquin Cubiella, Alberto Herreros-de-Tejada, Francesc Balaguer, Angel Lanas.

**Validation:** Romina Fernández-Poceiro, Rodrigo Jover.

**Writing – original draft:** Enrique Quintero.

**Writing – review & editing:** Natalia González-López, Antonio Z. Gimeno-Garcia, Luis Bujanda, Jesús Banales, Joaquin Cubiella, María Salve-Bouzo, Jesus Miguel Herrero-Rivas, Estela Cid-Delgado, Victoria Alvarez-Sanchez, Alejandro Ledo-Rodríguez, Maria Luisa de-Castro-Parga, Luciano Sanromán-Álvarez, Jose Santiago-Garcia, Alberto Herreros-de-Tejada, Teresa Ocaña-Bombardo, Francesc Balaguer, María Rodríguez-Soler,

Rodrigo Jover, Marta Ponce, Cristina Alvarez-Urturi, Xavier Bessa, Maria-Pilar Roncales, Federico Sopeña, Angel Lanas, David Nicolás-Pérez, Zaida Adrián-de-Ganzo, Marta Carrillo-Palau, Enrique González-Dávila.

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
