## [Editor Report · Decision Letter 0]

14 Mar 2023

Dear Dr Quintero, 

Thank you for submitting your manuscript entitled "Screening uptake of colonoscopy versus fecal immunochemical testing in first-degree relatives of patients with non-syndromic colorectal cancer: a multicentre, open-label, parallel-group, randomized trial (ParCoFit study)" for consideration by PLOS Medicine.

Your manuscript has now been evaluated by the PLOS Medicine editorial staff and I am writing to let you know that we would like to send your submission out for external peer review.

Please re-submit your manuscript within two working days, i.e. by Mar 16 2023 11:59PM.

Sincerely,

Philippa Dodd, MBBS MRCP PhD

PLOS Medicine

---

## [Decision Letter · Decision Letter 1]

25 Apr 2023

Dear Dr. Quintero,

Thank you very much for submitting your manuscript "Screening uptake of colonoscopy versus fecal immunochemical testing in first-degree relatives of patients with non-syndromic colorectal cancer: a multicentre, open-label, parallel-group, randomized trial (ParCoFit study)" (PMEDICINE-D-23-00676R1) for consideration at PLOS Medicine. 

Your paper was evaluated by a senior editor and discussed among all the editors here. It was also sent to independent reviewers, including a statistical reviewer. The reviews are appended at the bottom of this email and any accompanying reviewer attachments can be seen via the link below:

[LINK]

In light of these reviews, I am afraid that we will not be able to accept the manuscript for publication in the journal in its current form, but we would like to consider a revised version that addresses the reviewers' and editors' comments. Obviously we cannot make any decision about publication until we have seen the revised manuscript and your response, and we plan to seek re-review by one or more of the reviewers. 

We expect to receive your revised manuscript by May 16 2023 11:59PM. Please email us (plosmedicine@plos.org) if you have any questions or concerns.

We look forward to receiving your revised manuscript. 

Sincerely,

Philippa Dodd, MBBS MRCP PhD

PLOS Medicine

plosmedicine.org

GENERAL

Please respond to all editor and reviewer comments detailed below, in full

Please include line numbers starting at page line 1 “Abstract” and in continuous sequence thereafter

Thank you for reporting you study according to CONSORT. Please complete the CONSORT checklist and ensure that all components of CONSORT are present in the manuscript, including [how randomization was performed, allocation concealment, blinding of intervention, definition of lost to follow-up, power statement]. 

When completing the checklist, please use section and paragraph numbers, rather than page or line numbers as these often change in the event of publication. Please upload as supporting information.

The abstract is very nicely written. Thereafter, there are a number of grammatical errors which contribute to inaccessibility. Please check carefully and amend throughout. We suggest proof-reading by a native English speaker prior to resubmission.

*** The Editorial team agree that the striking feature of this study is the lack of uptake of screening by high-risk FDRs. We agree with the reviewers that this aspect should be a primary focus. As well as the specific items detailed below, you may wish to include additional revisions, considering the aforementioned. ***

DATA AVAILABILITY STATEMENT

Thank you for agreeing to make your data available upon request. The Data Availability Statement (DAS) requires revision. Study authors are not considered an appropriate contact for data requests. Please see the policy below and revise the availability statement accordingly.

For each data source used in your study: 

FUNDING STATEMENT

Please remove this statement from the main manuscript (page 2) and include only in the manuscript submission form upon resubmission. It will be compiled as metadata

ABSTRACT

It’s not entirely clear when reading the abstract that participants were randomised and then invited/consented to participate (this is most obvious form the flowchart, figure 1) Please amend for clarity. 

Suggest “randomized” perhaps, instead of “assigned”

Please define statistical abbreviations for the reader at first use – OR, CI and so on

Suggest reporting statistical information as follows, “(OR 1.12; 95% CI [0.84,1.49]; P=0.43)” for improved clarity – please note, the use of square parentheses and the use of commas to separate upper and lower bounds as hyphens can be confused with the reporting of negative values.

Thank you for including p values. Please report p as <0.001 and where higher as p=[exact p value]

Please ensure that all numerical values reported are consistent with those reported in the main manuscript text (see below)

Abstract conclusions:

Suggest “In this study…”

AUTHOR SUMMARY

Thank you for including an author summary, suggest some revisions as detailed below but further revisions may be warranted in view of comments above and reviewer comments below.

What did the researchers do and find? 

> We conducted a multicenter randomized controlled trial in Spain, including 870 asymptomatic first-degree relatives (FDR) of patients with non-syndromic CRC. Individuals were invited to participate in the setting of an opportunistic screening ** this term may be unfamiliar to some, please clarify what is meant by opportunistic/revise **

> FDR were randomly assigned (1:1) to one-time colonoscopy versus annual FIT during three consecutive years followed by a work-up colonoscopy in the case of a positive test. 

> Contrary to expectations, the rate of screening completion was similar in those assigned to FIT compared to colonoscopy screening (36% vs 34%). Consequently, the detection rate of advanced colorectal neoplasia was significantly lower in subjects receiving annual FIT than in those assigned to receive colonoscopy.

What do these findings mean? 

> The findings of this trial indicate that FIT does not increase colonoscopy screening completion rates in FDR of those diagnosed with colorectal cancer. 

> Improved education measures may help to improve awareness and screening uptake in this high-risk population. 

> Future studies are required to determine ways in which screening uptake can be improved for these individuals – perhaps through their inclusion in population-based screening programs or by offering options for screening.

Please include details of the study’s primary limitation as the final bullet point under “What do these findings mean? 

INTRODUCTION

There are a number of grammatical errors throughout, for example “…ten years before to the youngest case…” suggest “before the youngest…” perhaps?

And, “Nevertheless, clinical guidelines empirically recommend to perform colorectal cancer screening in these individuals separately from the average-risk programmed screening programs (opportunistic screening), being the most extended strategy colonoscopy every 5 years for higher-risk groups, starting at the age of 40 years or ten years before to the youngest case in their family [5-7].” – this is not very accessible and rather difficult to appreciate the strategy revise for clarity 

METHODS and RESULTS

See above comments under abstract – please ensure consistent and clear reporting of the randomisation process and its timing i.e. before invitation/consent as we understand things

Please justify the changes to secondary outcomes listed here compared to trial registration 

Page 8 – “If the index case agreed and met…” suggest reconsidering the use of the word ‘agreed’

Page 12 - “1, 870 relatives of 225 index cases” but in the abstract “1790 FDR of 460 index cases” please clarify. Please ensure that all numerical values reported throughout are consistent and accurate

Page 22 – “Overall, 309/383...” it’s not clear from the text how this denominator is derived and re: the numerator – does this include those who complied with FIT and those who complied with colonoscopy? If so, as per page 17-18 that would include 158 (note that table 7 details 162…) for FIT plus 147 for colonoscopy equivalent to 305 (as for the abstract)…please clarify/revise as necessary.

Please ensure all data reporting is clearly, accurately and consistently reported throughout all sections of the manuscript including in the tables and the figures.

*** The editorial team are in agreement with reviewer #4, please see below, that a comparison of those who participated Vs those who did not would be helpful - we note on page 28, the following “We could not contact these individuals as Spanish law does not allow registering of data of individuals that have not given previous informed consent.” And appreciate the limitations posed here. Are there any data available for these individuals? If so please include relevant comparison analyses. 

The discussion is heavily focussed on the superiority/inferiority of the individual screening approaches and the reviewers agree that this is a significant limitation. However, low uptake by participants in both arms is very striking and warrants further discussion***

TABLES

Please ensure tables are affiliated to an appropriate caption which clearly describes their content without the need to refer to the text. Please define any and all abbreviations including those used for statistical reporting. 

Throughout, please clearly indicate whether analyses are adjusted or unadjusted and in the footnote detail which factors are adjusted for.

Where adjusted analyses are presented, please also present the unadjusted analyses for comparison to help facilitate transparent data reporting. 

Please separate upper and lower bounds of 95% CIs with commas instead of hyphens as these can be confused with negative values

When reporting p values please also report 95% CIs for comparison

Table 1 – we agree with the statistical reviewer that the table can be moved to the supporting files

Table 5 – footnote d states “Advanced colorectal neoplasia was defined as an adenoma or serrated polyps measuring 10 mm or more in diameter, with villous architecture (>25%), high-grade dysplasia, in situ adenocarcinoma and invasive or CRC”. Please provide a breakdown into the different categories for this finding given the grouping applied.

FIGURES 

Figure 1 – includes a misspelling and a mark up to depict it, please amend, please ensure all numerical values are consistent with those reported in the main manuscript

DISCUSSION

Page 28 – please revise the first sentence for improved grammar/clarity.

Page 28 – “among the 383 that agreed to participate, this number isn’t mentioned in the abstract

Page 29 – PLOS Medicine does not permit “…(data not shown)…” please either remove reference to these data or include the data as supporting information.

*** As above – the discussion is heavily focussed on comparison of the 2 study arms and we agree that additional focus should be given to the low uptake the potential reasons for this and implications for practice/policy ***

SUPPORTING INFORMATION

Please include the study protocol document and analysis plan, with any amendments, as Supporting Information to be published with the manuscript if accepted.

Please include the CONSORT checklist

Comments from the reviewers:

Reviewer #1: General

The present pragmatic, multicenter randomized controlled trial compares, as the primary outcome, screening uptake of FIT with primary colonoscopy among first degree relatives (FDR) with a high-risk history of non-syndromic colorectal cancer (CRC). The detection of advanced colorectal neoplasia in the FIT groups vs. the colonoscopy group is the secondary outcome. The trial showed that FIT screening uptake was not superior to colonoscopy uptake. More advanced colorectal neoplasia was detected in the colonoscopy group. The authors conclude that new screening strategies should be investigated to increase screening uptake in this high-risk population.

In general, the paper is well written. The knowledge gap is clearly stated: "The screening uptake of FIT in individuals at high-risk for familial CRC has not been analyzed yet."

As CRC is a mayor health burden and FDR are at increased risk for developing the disease and colonoscopy uptake remains low in this population the research question is an important one. It can answer if FIT could be an appropriate method to get this high-risk population screened (however, the effectiveness of FIT on CRC incidence and mortality is unknown as long as we do not have results from FIT RCTs reporting on CRC incidence and mortality reduction). 

The authors follow CONSORT reporting guidelines throughout the manuscript and the study conforms to ethical guidelines.

The trial design is well suited to answer the research question. A pragmatic RCT with randomization before informed consent and intention-to-treat analysis is the gold standard to answer if an intervention could work in a real-life setting. In this way, the authors minimized the risk of bias. The trial population is well suited to test the hypothesis. The authors state the mayor limitation, the unsatisfactory participation rate in both groups, but I miss a more in-depth discussion of the impact of this limitation especially on the secondary endpoint. The primary endpoint is assessed and the conclusion may be valid due to sufficient statistical power. However, I am not sure if the result is generalizable to other countries with a different prevalence of colorectal neoplasia, different prevalence of risk factors and a differing attitude to CRC screening. With regard to the secondary endpoint the authors could have stated more clearly that only a very low number (29 individuals) had fulfilled 3 rounds of FIT testing and only 19 colonoscopies were performed in the FIT group. Detection rates may be based on chance and not be representative. 

Abstract:

Major issues:

- In the conclusion the authors state: "FIT did not improve the screening uptake of colonoscopy in individuals with a high-risk family history of non-syndromic colorectal cancer, resulting in less detection of advanced colorectal neoplasia."

- Comment: I am not sure if I understand that sentence completely. The outcome was to compare screening uptake with both strategies, but not "screening uptake of colonoscopy". As only some of the individuals participating in FIT will get a positive result, the colonoscopy uptake in the FIT group will be low, but the hope is to filter out those with advanced neoplasia and get those examined with colonoscopy. I suggest that the sentence should be rephrased to clearly state that FIT screening did not improve the screening acceptance compared to colonoscopy screening in this high-risk group.

What do these findings mean?

- In this chapter the above-mentioned example is repeated. The authors write: "The findings of this trial indicate that FIT does not have the capacity of increasing colonoscopy screening completion rates in the non-syndromic familial colorectal cancer population.

- Comment: It was not expected that FIT could increase colonoscopy completion. The hope was that FIT could increase screening completion.

Introduction

The introduction is well written and summarizes the literature clearly. It explains the high-risk for FDR to develop CRC themselves but the uncertainty regarding the benefit of intensive colonoscopy surveillance. It describes the problem of low compliance with colonoscopy screening in this group and explains that repeated annual FIT may be an appropriate screening method for this group due to good diagnostic yield for CRC and advanced adenoma. The introduction leads to the knowledge gap: "screening uptake of FIT in this population has not been analyzed yet." 

Major issues:

- The authors state that: "this study was designed to test the hypothesis that uptake of FIT screening followed by a work-up colonoscopy in the case of a positive test is superior to colonoscopy screening with an equivalent effect on colorectal cancer prevention in the population with high familial risk."

- Comment: This study was designed to compare uptake rates and detection rates. The effect on colorectal cancer prevention depends on more than that, e.g. adequate removal of neoplasia. I suggest being a little more cautious with the phrasing here.

Methods

The methods chapter is clearly and understandably described. 

Minor issues:

General

- Comment: I am surprised about the low participation rate and wonder if it is possible that the invitation to the trial came "too late". Is it possible that a relatively large proportion of FDR already contacted health providers to get a colonoscopy done after the diagnosis of the index case and before invitation to screening? This possible explanation for low uptake could also have been elaborated in the discussion.

- The authors write that "Sedation and colon cleansing were performed as previously described."

Comment: Sedation is not described in the mentioned reference (19).

- The authors state that: Patients with an incomplete colonoscopy due to any technical difficulty that impeded the exploration of the cecum had to be evaluated by CT colonography or colonic capsule endoscopy.

Comment: It is unusual to offer colonic capsule endoscopy, a method lacking evidence for the efficacy for colorectal neoplasia detection.

- Severe complications are defined as immediate and delayed postpolypectomy haemorrhage and intestinal perforation. I suggest that the authors describe how haemorrhage and perforation were defined. Need for transfusion, fall in Hb, free air on CT and so on.

- It is detailed that "They were notified to deliver it to the laboratory within 7 days. Individuals with ≥10 μg Hb/g faeces were invited to undergo colonoscopy." 

Comment: This timeframe seems to be very short and may partially explain low participation. What happened if no sample was delivered within 10 days? Was it possible to send the sample later? Was a reminder send? This could be specified here and discussed in the discussion as well.

- Arm and group are used interchangeably throughout the manuscript. I propose to consequently use either group (preferably) or arm. 

- In the chapter called "Follow up", I suggest moving up the definition of interval colorectal cancer. Now, it is first described how interval cancers are identified before, in the last sentence of the chapter, the definition is stated. 

- "Statistical analysis and sample size calculation": Wouldn't it be more appropriate to use age as a continuous variable or in categories with one year increase for multivariable logistic regression analysis instead of categorizing in two categories according to the median age of participants?

Results

The results are in general clearly presented and tables and figures are easy to read and understand.

- Figure 1: 

o typing error: Syndromic hereditary. 

o Number of non-compliant individuals do not match the number in the text (74 individuals in total) and Table 7: 49 or 48 individuals in the colonoscopy group (+ 25 in the FIT group)?

- Table 5:

o Detection rate of serrated adenoma is stated. According to the method section, serrated polyps are classified according the WHO guidelines. The current guidelines group serrated class polyps into hyperplastic polyps, sessile serrated lesion with and without dysplasia and traditional serrated adenomas. The last two groups are considered as possible precancerous neoplasia. It is not clear what is meant by "serrated adenoma" and should be rephrased according to the guidelines and be explained in the method section. It could be reasonable to report on advanced serrated lesions (serrated polyps >=10mm and/or with dysplasia).

- Table 7: In the footnote, unplanned FIT screening is defined, but not unplanned colonoscopy screening (c is displaced)

Discussion

The discussion is well written and places the trial results in the context of the literature. The authors state both strengths and limitations of the trial in an adequate way. I miss as mentioned earlier a more thorough discussion regarding the impact on low participation on the validity of this trial regarding detection rates and generalizability of this trial. The low participation must have been surprisingly for the authors as well. Repeated FIT screening in programmatic screening has in Europe an overall participation of 50%, but in trial settings a higher participation has been observed (65-70%). Why is the participation in a group informed about an increased risk for CRC so low? The authors could have elaborated on other possible mechanisms. Is it conceivable that those who would have been compliant with recommendations already have been screened before invitation to the trial and a large proportion of those eligible for the trial were non-compliers?

Inclusion from different regions and centres is an important strength of the trial but in the face of low participation I am in doubt if the results can be extrapolated to other countries with a different screening adherence.

The effectiveness of screening depends on the performance of the test, the performance of the programme and the participation. FIT screening needs to be repeated to reach sensitivity of colonoscopy. Low participation and a very low number of participants who participated in several rounds of FIT screening weaken the validity of this trial. 

Minor issues

- The authors state as earlier: "The current study shows that in Spain, a country with universal public health care and more than a decade of experience with a FIT-based nationwide screening program, repeated FIT did not have the capacity to improve the screening uptake of colonoscopy in the familial-risk population." and in the conclusion in the end: "In conclusion, this randomized controlled trial indicates that in the setting of an opportunistic screening, annual FIT does not increase the screening uptake of colonoscopy screening in

FDR at high risk of developing colorectal cancer, resulting in a significantly lower detection rate of advanced colorectal neoplasia.

Comment: I believe what they want to say is that FIT has not the capacity to improve the screening uptake compared to colonoscopy… as FIT unlikely can improve uptake of colonoscopy.

- The authors write that: "Traditionally, FDR of patients with colorectal cancer have been excluded from nationwide screening programs because they are considered candidates for straightforward colonoscopy.

Comment: I am not sure if that statement is correct. Probably Spain excludes FDRs from programmatic screening. Do they have a register to identify those who have a FDR with CRC? 

FDRs are not excluded from the British bowel screening programme, and they are not excluded from the nationwide screening programme in Norway, Danmark and Sweden. FDRs to individuals with CRC in these countries are recommended to have a colonoscopy but are still invited to the screening programmes. 

I definitely agree with the authors that these individuals at high risk should be involved in organized screening programmes and that increased awareness of and information about risk of CRC is crucial to increase screening adherence.

- Typing error, double word: "In line with this finding, a recent a meta-analysis of four controlled trials [31] showed that tailored communication based on repeated contact combined with written and verbal information increased the participation rate in colonoscopy screening by about twofold."

Reviewer #2: Statistical review

This paper reports a randomised controlled trial comparing screening of first degree relatives of individuals with colorectal cancer. The authors showed there was no significant difference in screening uptake, although there was in the proportion of participants who had cancer detected.

I had some comments on the paper which I have provided below.

1. I did not see the trial protocol provided as supplementary material.

2. Abstract: I would recommend the p-value for the detection rate is reported more precisely than '<0.01'.

3. Page 8 "and was checked for compliance with the CONSORT (Consolidated Standards of Reporting Trials) checklist" - I would rephrase this as 'reported according to CONSORT'.

4. Page 8 "proper randomization sequence and allocation concealment were ensured" - could more be said about this (i.e. how concealment was ensured)? Also, was randomisation simple randomisation or using blocks/strata?

5. Outcomes: the secondary outcomes here do not match the ones listed on the clinicaltrials.gov registration (detection rate is not mentioned and costs and QALYs are mentioned). I would recommend changes in outcomes are mentioned.

6. Statistical analysis: for the detection rate outcome, and the denominator be clarified: is it the number of subjects enrolled in the arm or the number who took up the screening? I think the former would be preferable as it preserves the randomisation. I see later in the results it is the former.

7. Statistical analysis: It is good that the interim analysis is reported clearly. I am not sure that all the metrics/table 1 are required though. Table 1 could perhaps be better as supplementary material. I have not come across the futility index before, is it just 100-conditional power, or is that a coincidence?

8. Page 18 "was similar" would be better as "was not significantly different"

9. Page 20 - I did not follow why the results in the text for the advanced colorectal neoplasia were different to those in table 5 in terms of CI and p-value (the caption says that adjustment was made).

10. Table 6: Please report p-values more precisely than <0.01, <0.05.

James Wason

Reviewer #3: The paper is reporting the results of a RCT designed to compare uptake and neoplasia yield of annual low-cut-off FIT screening and of a single colonoscopy screening in a population at high familial risk, without hereditary syndromes.

The management of these subjects is showing a wide variability and there is uncertainty about the best way to reach them as well as about the best screening strategy.

The trial was well designed and conducted and its results are confirming that management of these subjects poses several challenges 

The authors found a low participation also to non-invasive screening, in this group.

However, as they mention, the experimental setting might have influenced the response, as , due to the randomization approach, about 35% of subjects in the same family were randomized to different trial arms, which could negatively influence participation.

Also about 20% of those invited did not follow the planned protocol and a non- negligible number of potentially eligible FDRs were excluded as they had already undergone screening.

So the observed response rate, likely refer to the sub-group of subjects less aware of the problem or having negative or fatalistic attitudes. The opportunistic setting represented also a barrier to attendance. These barriers apply to both arms, but the conclusions about participation might be integrated referring to these potential determinants of low compliance.

The authors might report, to provide the full picture, information about the type of test performed before the offer of recruitment and outside the trial.

One important conclusion, mentioned also by the authors, which could be stressed, is that, given these results, these subjects at high familial risk should not be excluded from invitation in population based screening programs.

Minor comment

In the methods the authors mention that they collected information about interval cancer and mortality, but the results are not mentioning these outcomes

Reviewer #4: This is a potentially important study, but it was hampered by low uptake of the screening invitation in both arms. It is hard to draw conclusions based on these results, and I worry that publishing it in its current form will lead to misunderstanding of the role of FIT in screening. The key issue was a low uptake of the invitation, regardless of the screening strategy. 

A more interesting and informative analysis would be to more fully elucidate the characteristics of people who participated in the trial vs those who did not. Did they differ by the age or sex of the index case? Did they differ according to the stage at diagnosis or the location of the tumor? 

Do they know whether the index cases actually told their family members about it? Clearly these folks would have benefited by direct contact from study participants. 

Reviewer #5: This paper addresses a crucial concerning the adherence to FIT screening in individuals at high-risk for familial colorectal cancer. This study found that the screening uptake to FIT was not higher than the uptake to colonoscopy, resulting in a lower detection rate of advanced colorectal neoplasia in FIT group than in the colonoscopy group. However, there are concerns regarding a couple of issues in the study. 

Major issue:

1. The inclusion of screen-naïve FDR may have introduced selection bias as these individuals have not yet participated in any colorectal cancer screening program. It is possible that these individuals are less likely to participate in screening at all. This low participation rate is evident as only 40% of all individuals agreed to participate in the study. These issues should be discussed in the paper's discussion section. Additionally, it is worth noting that the participation rate is higher in the FIT group than in the colonoscopy group if calculated based on the number of individuals who agreed to participate (158/187 vs 147/196). This finding is interesting and should be discussed in the paper. Furthermore, a large number of unplanned FIT was found in the colonoscopy group. Were those FITs performed as screening tool, or because of symptoms? If performed as screening tool, that may imply a preference for FIT in at least part of the individuals. This should also be discussed in the paper. 

Minor issues:

1. In the introduction, it is mentioned that less than 50% of FDR of patients with colorectal cancer undergo colonoscopy screening. Please clarify this type of screening. Is it 10-yearly colonoscopy? 

2. How well is the registration of colonoscopies and FIT outside of this study? Are FITs performed via the screening program included in this study? And why is a FIT performed as a screening tool only included as unplanned FIT in the colonoscopy group? How is a FIT performed via the screening program in the FIT group than included?

3. The paper does not provide a clear indication whether colonoscopies performed in the context of screening are always free of charge in Spain. If they are not, the screening uptake for colonoscopy may be lower than reported in the study. Furthermore, it is unclear whether the colonoscopies performed after a positive FIT result were free of charge. These factors should be taken into account and discussed in the paper.

4. How many individuals in both groups did not attend the initial appointment, but were considered as eligible? Did this number differ between groups? 

5. The reason for non-compliance needs to be added to the flowchart, as this is currently unclear. In addition, 1 person is missing from the flowchart in the colonoscopy group. 147 underwent screening in that group, and 48 were non-compliant. That makes a total of 195, but 196 individuals attended the appointment. 

6. The tables containing the multivariable logistic regression models require further elaboration. The results of a univariable analysis should be included. Additionally, the statement in the paper that "after adjusting for demographic characteristics, we found that screening uptake was similar in both groups" is not reflected in Table 4. Based on the results in table 4, you can make a statement like: screening strategy, gender, age and participating centers were not significantly associated with screening uptake. Please change the table accordingly to make a statement like: screening uptake was not significantly different between groups, also when adjusting for other variables. In addition, why were only those 3 demographic variables included in the multivariable analysis, and not, e.g. inclusion criteria (one index case < 60 etc.)? I am also curious to see the effect of that variable. 

7. In Table 7, the number of individuals who are compliers with the assigned strategy needs to be accompanied by a definition of what constitutes compliance. As this is not clear in the paper, especially as these numbers are not presented in the flowchart. For FIT, is someone a complier after completing 1, 2 or 3 FITs?

8. In Table 7, the number of individuals under post-polypectomy surveillance is much higher in the colonoscopy group. How is that possible? And should those individuals not be excluded from the analyses, as post-polypectomy surveillance is different from screening?

9. Please check the numbers in Table 7, as the numbers in the FIT group do not add up to the total number of compliant/non-compliant. 

10. Table 7 needs to clarify when individuals are deemed compliant or non-compliant. Is it only when they did not attend the initial appointment? Individuals in the non-compliant group underwent some type of screening, but it is not clear from the text/table.

11. The discussion states that 'only 34.9% of participants were randomly assigned to different strategies within the same family'. This is a substantial percentage and may have influence the participation rate. The paper should provide further hypotheses about this factor.

[LINK]

---

## [Decision Letter · Decision Letter 2]

26 Jul 2023

Dear Dr. Quintero,

Thank you very much for submitting your manuscript "Screening uptake of colonoscopy versus fecal immunochemical testing in first-degree relatives of patients with non-syndromic colorectal cancer: a multicenter, open-label, parallel-group, randomized trial (ParCoFit study)" (PMEDICINE-D-23-00676R2) for consideration at PLOS Medicine. 

Your paper was evaluated by a senior editor and discussed among all the editors here. It was also discussed with an academic editor with relevant expertise, and reviewed again by the reviewers, including the statistical reviewer. The reviews are appended at the bottom of this email and any accompanying reviewer attachments can be seen via the link below:

[LINK]

In light of these reviews, I am afraid that we will not be able to accept the manuscript for publication in the journal in its current form, but we would like to consider a revised version that addresses the reviewers' and editors' comments. Obviously we cannot make any decision about publication until we have seen the revised manuscript and your response, and we plan to seek re-review by one or more of the reviewers. 

We expect to receive your revised manuscript by Aug 09 2023 11:59PM. Please email us (plosmedicine@plos.org) if you have any questions or concerns.

We look forward to receiving your revised manuscript. 

Sincerely,

Philippa Dodd, MBBS MRCP PhD

PLOS Medicine

plosmedicine.org

GENERAL 

Thank your detailed and considered responses to previous editor and reviewer comments. Please respond to all editor and reviewer comments detailed below in full.

Please include line numbers in your revised version, beginning at 1 and in continuous sequence thereafter. 

*** The reviewers suggest further analyses, please see below, which have the potential to alter the interpretation of your data and for this reason we have requested a 2nd major revision. ***

ABSTRACT

Background:

Suggest ‘we investigated’ instead of ‘we analyzed’ 

Should the word ‘prevention’ be replaced with ‘detection’ in the final line? Similarly in the author summary (please see below).

Methods and findings:

Please remove the word ‘prospective’ which better describes a cohort study as opposed to a randomized trial

Primary outcome – please detail ‘colonoscopy and FIT’ instead of ‘both strategies’

Please use lower case p to depict the p values.

Conclusions:

Suggest, ‘In this study, compared to colonoscopy, FIT screening did not improve screening uptake by individuals at high risk of CRC, resulting in less detection of advanced colorectal neoplasia. Further studies are needed to assess how screening uptake could be improved in this high-risk group, including by inclusion in population-based screening programs.

AUTHOR SUMMARY

Why was this study done?

Suggest combining points 1 and 2 to read as follows:

‘The risk of colorectal cancer (CRC) is three to four times higher in first degree relative (FDR) of patients with non-syndromic colorectal cancer. These individuals are considered candidates for colonoscopy-based screening starting at 40 years of age, but this approach is associated with a suboptimal acceptance rate of approximately 50%.’

Please include the abbreviation ‘(CRC)’ before first use later

Point 3 – suggest ‘acceptance of’

Point 4 – should ‘prevention’ be ‘detection’ here?

What did the researchers do and find?

Point 1 - please split into 2 bullet points at sentence beginning ‘FDR…’

Point 2 – suggest removing ‘contrary to expectations’ and making the remaining sentences into 2 points as follows:

‘The rate of screening completion was similar in the group assigned to FIT and the group assigned to colonoscopy screening (36% vs 34% respectively).

‘The detection rate of advanced colorectal neoplasia was significantly lower in subjects receiving annual FIT than in those assigned to receive one-time colonoscopy’.

What do these findings mean?

In point 1 you state, ‘The findings of this trial indicate that FIT does not have the capacity of increasing colonoscopy screening completion rates in the non-syndromic familial colorectal cancer population.’ But this is inconsistent with the defined hypothesis, ‘that uptake of FIT screening is superior to that of colonoscopy screening in this population’. Suggest rephrasing for clarity here and to ensure consistency throughout.

Point 2 – please remove the word ‘Meanwhile’ and make sentence beginning ‘Future studies are…’ a separate bullet point. 

INTRODUCTION

Sentence beginning, ‘Recently, a large prospective study…’ suggest making a new paragraph 

Page 6 final paragraph – suggest avoiding repeated use of the word ‘recently’

Final sentence - ‘…test the hypothesis that uptake of FIT screening followed by a work-up colonoscopy in the case of a positive test is superior to colonoscopy screening with an equivalent effect on colorectal cancer detection in the population with high familial risk’ this is again different to the abstract and the author summary. We think that you are trying to say ‘uptake of FIT screening followed by a work-up colonoscopy…is superior to uptake of one-time colonoscopy screening…’ as we understand things. Suggest revising throughout all sections to ensure that the hypothesis is clearly and consistently defined for the reader.

METHODS and RESULTS

Please define the acronym ‘CONSORT’ for the reader, as in your previous version, my apologies for not making this clear previously.

Please also see reviewer comments detailed below regarding additional analyses required, which we agree with.

FIGURES

Figure 1 – please ensure all abbreviations are defined for the reader in an appropriate footnote/caption.

TABLES

The terms ‘male and female gender’ are used incorrectly (table 1, for example). ‘Gender’ should be replaced with ‘Sex’. Alternatively, simply detailing ‘Male n (%)’ and ‘Female n (%)’ would suffice.

Please check and revise throughout all tables including those in the supporting files as appropriate.

DISCUSSION 

Line 1 – suggest removing the word ‘pragmatic’.

REFERENCES

For in-text reference callouts please remove the space between different citations to read as follows, [1,3,6] (as opposed to [1, 3, 6]). Please check and amend throughout all sub-sections of the manuscript and supporting information.

SUPPORTING INFORMATION 

CONSORT checklist – please remove the numbers 1 and 2 from the first 2 lines (as these refer to page numbers). ‘Title’ and ‘Abstract’ will suffice.

STATISTICAL ANALYSIS PLAN – please ensure that the reference formatting follows our guidance, as for the main manuscript, including in-text reference callouts. 

Please note that we cannot publish copyright symbols such as ©, ®, or ™ (i.e. OC-Sensor ® kit) please revise throughout where relevant including the main manuscript.

Comments from the reviewers:

Reviewer #1: Thank you for this revised manuscript. Many of my comments are answered in a satisfactory way. Thank you also for clarifying answers that did not result in changing the manuscript. 

I just want to mention that the Nordicc trial is a primary colonoscopy trial and not a FIT trial (as you mention a low uptake in FIT screening in Norway in one answer). In the Norwegian Pilot for the national screening programme the participation rate was 68% with 3 rounds of FIT (Randel KR, Schult AL, Botteri E. et al. Colorectal cancer screening with repeated fecal immunochemical test versus sigmoidoscopy: baseline results from a randomized trial. Gastroenterology 2021; 160: 1085-1096 e1085). 

I still have some comments to your revised manuscript:

Abstract and author summary:

-You use the term advanced colorectal neoplasia without defining it. It is defined later in the methods but should be defined here as well.

Author summary 

-"What do these findings meet", I see that the statement "What do these findings mean" is changed, but I still find it not clear. "The findings of this trial indicate that FIT does not have the capacity of increasing the acceptance of colonoscopy screening in the non-syndromic familial colorectal cancer population. " In my opinion this should be rephrased: "The findings of this trial indicate that FIT does not have the capacity of increasing the acceptance of screening in the non-syndromic familial colorectal cancer population." 

Methods/Study procedures: 

- Grammatical error: "Colon cleansing were performed as previously described [19]." must be changed to "was" performed. 

-"They were notified to deliver it to the laboratory within 14 days." I think it should be clarified that what you mean is that they were notified to deliver it within 14 days after taking the sample and not within 14 days after receiving the kit. 

-Grammatical error: "The individuals who did not deliver the test on time were contacted by telephone for offer them a new test" should be "to" offer

-"We considered severe post-polypectomy bleeding if prevented the conclusion of the procedure o transfusion was required." Is what you mean: that the completion of the procedure was prevented or blood transfusion was required? What about hospitalization or repeated endoscopy? That was not a criteria? 

Results:

- Table 5: please check the WHO definition for sessile serrated lesions. Now you changed the changed the term but it is not correct. You wrote in your answer that there was no dysplasia and therefor there were no advanced serrated lesions. But you write in the table sessile serrated adenoma. This term dose not exist. All adenomas have dysplasia. I think what you mean is non-advanced sessile serrated lesions. Is it correct that no traditional serrated adenomas were detected (all TSA have dysplasia).

General:

-there are some spelling errors, double words, missing commas throughout the manuscript.

Reviewer #2: Thanks to the authors for addressing my previous comments. I just wanted to highlight that describing the randomisation as 'simple randomisation' and 'stratified by centre' is slightly contradictory. I assume that stratified block randomisation was used? Simple randomisation would just be randomly allocating each individual with 50% probability.

Reviewer #5: Reviewer reply to PMEDICINE-D-23-00676_R2

The authors have improved the manuscript and incorporated most of my comments. However, I do have one follow-up comment that needs to be addressed before publication. 

1. The authors now state in the rebuttal as well as in the Discussion: "Consequently, some individuals had a different screening strategy assigned in the same family, which could have led to refusal to participate in some cases. In fact, the assignment of different strategies to members of the same family was an independent factor for low participation in the logistic regression analysis. However, the rate of subjects who were assigned to different strategies in the same family was similar in the FIT group (51.3%), and in the colonoscopy group (48.7%), suggesting that this condition affected both groups equally."

This statement cannot be made based on these results only. You can only state that the rate of subjects who were assigned to difference strategies in the same family was similar in both groups. However, it may be possible that, for example, those who were assigned to FIT, were less likely to participate with FIT, because family members were assigned to colonoscopy (and thereby being less comfortable with less invasive surveillance). 

In order to make this statement, the effect of different strategies assigned should be analysed separately for the FIT and the colonoscopy group: thus, what is the screening uptake in the FIT group stratified for those assigned same and different strategy, and the uptake in the colonoscopy group stratified for those assigned same and different strategy. If those results show similar results in participation rates, I agree with the statement. However, if those results show differences in participation rates, it should be clearly stated that this is likely to have affected the results, and make definitive conclusions about the difference in participation between annual FIT and colonoscopy complicated.

[LINK]

---

## [Decision Letter · Decision Letter 3]

1 Sep 2023

Dear Dr. Quintero,

Thank you very much for re-submitting your manuscript "Screening uptake of colonoscopy versus fecal immunochemical testing in first-degree relatives of patients with non-syndromic colorectal cancer: a multicenter, open-label, parallel-group, randomized trial (ParCoFit study)" (PMEDICINE-D-23-00676R3) for review by PLOS Medicine.

I have discussed the paper with my colleagues and it was also seen again by 3 reviewers. I am pleased to say that provided the remaining editorial and production issues are dealt with we are planning to accept the paper for publication in the journal.

[LINK]

We look forward to receiving the revised manuscript by Sep 08 2023 11:59PM.   

Sincerely,

Philippa Dodd, MBBS MRCP PhD

PLOS Medicine

plosmedicine.org

Requests from Editors:

GENERAL

Thank you for your considered and detailed responses to previous editor and reviewer comments. Please see below for further comments which we require you address prior to publication.

COMPETING INTERESTS

Please ensure that competing interests have been declared as per the PLOS policy, which can be seen here:

https://journals.plos.org/plosmedicine/s/competing-interests

For authors with ties to industry, please indicate whether any of the interests has a financial stake in the results of the current study.

ABSTRACT

In the last sentence of the Abstract Methods and Findings section, please describe the main limitation(s) of the study's methodology.

AUTHOR SUMMARY

Line 74 – suggest ‘is that it was…’ also suggest expanding (very briefly) to detail that this impedes understanding of the very low screening uptake across both study arms. Perhaps, ‘The main limitation of this study is that it was not possible to collect information on eligible individuals who declined to participate thus impeding our understanding behind low screening uptake in this population.’ Or similar.

INTRODUCTION

Lin 99 – please remove the underscore preceding the opening bracket which appears to be a typographical error.

METHODS and RESULTS

Line 175 – please replace the term, ‘OC-Sensor kit’ with ‘quantitative FIT (Faecal Immunochemical Test)’ as the former is a trademark label. Please check and amend throughout. You could elaborate to describe the test ‘designed to detect occult human haemoglobin in human stool’ or similar perhaps.

Line 222 – suggest ‘were’ instead of ‘was’.

TABLES

Table 1 – is there a way to reduce the space between n numbers and percentages in columns 2 and 3? This would improve accessibility to the reader.

Table 2 – line numbers overwrite the data in the final column, bottom 4 rows.

Table 5 – final row, penultimate column, please replace the hyphen with a comma.

SUPPORTING INFORMATION

The supporting information name and number are required in a caption, and we highly recommend including a one-line title as well. You may also include a legend in your caption, but it is not required. 

In the published article, supporting information files are accessed only through a hyperlink attached to the captions. For this reason, you must list captions at the end of your manuscript file. You may include a caption within the supporting information file itself, as long as that caption is also provided in the manuscript file. Do not submit a separate caption file.

Please revise the ‘Supplementary Information Files’ list on page 46 of the manuscript PDF to detail instead only the relevant titles/captions for the figures/tables.

STUDY PROTOCOL

* Page 4: as above, please replace, ‘(OC-Sensor™)’ with ‘quantitative FIT (Faecal Immunochemical Test), designed to detect occult human haemoglobin in human stool’. Please check and amend throughout as the former is trademark label. Please check and amend throughout.

* Please replace hyphens with commas when reporting 95% CIs as the former can be confused with reporting of negative values. I note that on occasion the word 'to' is also used. Please check and amend throughout for consistency and clarity.

* Please ensure that for in-text reference callouts, citations are placed in square brackets as for the main manuscript. Please check and amend throughout all supporting files.

* Page 5, para 3: please replace the full stop with a comma when reporting citations, ‘[10,11]’ (as opposed to ‘(10.11)’).

* References – please ensure that the reference formatting follows our guidance as for the main manuscript. Please see here for further details https://journals.plos.org/plosmedicine/s/submission-guidelines#loc-references. Please list up to but no more than 6 author names followed by et al in the event that more than 6 authors contribute to a study.

SOCIAL MEDIA

To help us extend the reach of your research, please detail any Twitter handles you wish to be included when we tweet this paper (including your own, your coauthors’, your institution, funder, or lab) in the manuscript submission form when you re-submit the manuscript.

Comments from Reviewers:

Reviewer #1: Thanks to the authors for addressing my previous comments. 

Reviewer #5: Thanks to the authors for performing the requested analysis. These results indeed clearly show that, although the overall screening uptake was signficantly lower in those assigned different strategies (which the authors already state based on the logistic regression analysis), but that the screening uptake of colonoscopy did not significantly differ from the screening uptake of FIT in each of the two categories.

[LINK]

---

## [Editor Report · Decision Letter 4]

15 Sep 2023

Dear Dr Quintero, 

On behalf of my colleagues and the Academic Editor, Dr. Aadel Chaudhuri, I am pleased to inform you that we have agreed to publish your manuscript "Screening uptake of colonoscopy versus fecal immunochemical testing in first-degree relatives of patients with non-syndromic colorectal cancer: a multicenter, open-label, parallel-group, randomized trial (ParCoFit study)" (PMEDICINE-D-23-00676R4) in PLOS Medicine.

Prior to publication we require that you address the following:

* Please include a copy of the full trial protocol document in both its original form and in a version that is translated into English. We cannot publish your manuscript without this.

* S2 Table - please define OR and CI in the footnote

PRESS

Best wishes,

Pippa 

Philippa Dodd, MBBS MRCP PhD 

PLOS Medicine